# Establishment and characteristics analysis of a crop-drought vulnerability curve: a case study of European winter wheat

Yanshen Wu[1,2], Hao Guo[3], Anyu Zhang[1,2], Jing'ai Wang[1,2,4]

[1]School of Geography, Faculty of Geographical Science, Beijing Normal University, Beijing 100875, China;
[2]Key Laboratory of Environmental Change and Natural Disaster, MOE, Beijing Normal University, Beijing 100875, China
[3]College of Geography and Environmental Sciences, Zhejiang Normal University, Jinhua 321004, China
[4]College of Biologic and Geographic Sciences, Qinghai Normal University, Xining 810008, China

*Correspondence to*: Jing'ai Wang (jwang@bnu.edu.cn)

**Abstract.** As an essential component of drought risk, crop-drought vulnerability refers to the degree of the adverse response of a crop to a drought event. Different drought intensities and environments can cause significant differences in crop yield losses. Therefore, quantifying drought vulnerability and then identifying its spatial characteristics will help understand vulnerability and develop risk-reduction strategies. We select the European winter wheat growing area as the study area and 0.5°×0.5° grids as the basic assessment units. Winter wheat drought vulnerability curves are established based on the Erosion-Productivity Impact Calculator model simulation. Their loss change and loss extent characteristics are quantitatively analysed by the key points and cumulative loss rate, respectively, and are then synthetically identified VIA K-means clustering. The results show the following. (1) The regional yield loss rate starts to rapidly increase from 0.13 when the drought index reaches 0.18 and then converts to a relatively stable stage with the value of 0.74 when the drought index reaches 0.66. (2) In contrast to the Pod Plain, the stage transitions of the vulnerability curve lags behind in the southern mountain area, indicating a stronger tolerance to drought. (3) According to the loss characteristics during the initial, development and attenuation stages, the vulnerability curves can be divided into five clusters, namely, Low-Low-Low, Low-Low-Medium, Medium-Medium-Medium, High-High-High and Low-Medium-High loss types, corresponding to the spatial distribution from low latitude to high latitude and from mountain to plain. The paper provides ideas for the study of the impact of environment on vulnerability, and for the possible application of vulnerability curve in the context of climate change.

## 1 Introduction

Drought is a widespread natural disaster causing the largest agricultural losses in the world. More than one-half of the earth is susceptible to drought, including nearly all of the major agricultural areas (Kogan, 1997). Under the context of climate change and globalization, drought will pose a threat to future food security. How to assess and manage agricultural drought risks has become a focus of the world (Reid et al., 2006;Li et al., 2009;Mishra and Singh, 2010). As vulnerability is a key factor in determining risk, drought vulnerability assessment is an important foundation for drought risk assessment and management (Knutson C, 1998;Zhang et al., 2015).

Crop drought vulnerability assessment focuses on crops, particularly the biophysical factors closely related to crop growth processes (Tánago et al., 2015; Wu et al., 2017), describing the damage to crops caused by different intensity hits. At present, crop drought vulnerability assessment methods mainly include the following three aspects.

(1) Calculation of the comprehensive vulnerability index based on selected relevant indicators. Some of these studies encompass recognition of the factors influencing drought vulnerability, construction of vulnerability indicators from physiographic, climatic and hydrologic aspects, assignment of their weights and calculation of a comprehensive index (Wilhelmi and Wilhite, 2002;Shahid and Behrawan, 2008;Jain et al., 2014). For example, Pandey et al.(2010) identified seven influence indicators, such as

watershed geography, soil types, water availability and so on, graded each of them and then added them up to obtain the drought vulnerability index of the Sonar basin in the Madhya Pradesh. Some of these studies are based on the components of vulnerability, construct sensitivity and exposure indicators, and combine them to form a vulnerability index (O'Brien et al., 2004;Antwi-Agyei et al., 2012;Tánago et al., 2015). For example, Simelton et al. (2009) used the crop failure index to characterize

sensitivity, the drought index to characterize exposure, and the ratio of the two to characterize crop drought vulnerability, and then discussed the correlation between drought vulnerability and socio-economic characteristics in China. Although this method cannot predict the loss quantitatively and has certain subjectivity and uncertainty in index system construction and weights determination, affected by the difficulty of testing and verification, it is able to express the relative level of vulnerability between

regions, providing potential ways to disaster mitigation for decision makers, and providing a strong reference for the establishment of quantitative vulnerability relationships (Wilhelmi and Wilhite, 2002;Simelton et al., 2009;Wu et al., 2010).

(2) Quantitative research on vulnerability based on historical statistics and meteorological observations. This method mainly uses meteorological observation data and historical statistical data to build a

quantitative relationship between disaster intensity and historical disaster loss (Lobell and Burke, 2008;Hlavinka et al., 2009;Rowhani et al., 2011). Fishman (2016) used Indian daily rainfall and statistical yield data from 1970 to 2003 to analyse the relationship between precipitation variability and major crop yields. Jayanthi et al. (2014) used satellite rainfall based-water requirement satisfaction index and historical yield loss rate as regression indicators to develop a maize drought vulnerability model in Kenya,

Malawi and Mozambique. Xu et al. (2013) selected consecutive rainless days as the drought index, converted drought affected area into the drought-induced yield loss rate, and then established vulnerability curves of corn, wheat and rice in the monsoon region of east China based on the daily precipitation data and historical disaster data. Such a method explores how crop yield loss varies with disaster intensity, but is easily affected by the availability and quality of disaster loss data, therefore,

having difficulties in high-resolution vulnerability assessment and spatial analysis.

(3) Quantitative research on vulnerability curves based on field experiments and crop model simulations. This method generally conducts field experiments or crop growth model simulations by artificially setting up different disaster intensity scenarios, and then fitting cooperative vulnerability curves from the perspective of a crop disaster-causing mechanism. Pan et al. (2017) conducted field experiments by

artificially controlling soil water content at the Huanghua experimental site in Hebei, China. Based on

the experimental data of maize growth under different drought intensity, the physical drought vulnerability curves of the five growth stages were constructed. Yin et al. (2014) used the Erosion-Productivity Impact Calculator (EPIC) model to obtain drought index and yield loss rates, and constructed drought vulnerability curves for maize in 35 regions of the world. Kamali et al. (2018b) used the precipitation and EPIC-simulated maize yield data to describe the crop sensitivity and exposure indexes to drought, respectively, and linked the two indexes using a power curve fitted to describe the physical vulnerability of sub-Saharan African countries. This method provides a new research idea and perspectives for vulnerability quantitative assessment based on the crop growth mechanism. Additionally, the crop model can quantitatively predict the crop growth and yield formation process in a specific environment, with lower cost than the field experiments and fewer limitations in historical disaster statistical sample or spatial accuracy, which is conducive to high-precision quantitative research on crop vulnerability (Palosuo et al., 2011;Challinor et al., 2009). However, it is worth mentioning that as infinite dimensional data (James and Sugar, 2003), the vulnerability curve used in this method is difficult to perform spatial analysis directly like the vulnerability index, which leads to its main application in risk assessment field with insufficient vulnerability information mining.

Actually, there are spatial differences in crop drought vulnerability affected by factors such as the natural environment and crop variety (IPCC, 2012, 2014). Analysing and mapping the spatial differences based on a quantitative assessment can better help identify the vulnerability distribution and local mitigation-oriented drought management. Therefore, this paper aims to exploring the vulnerability curve feature extraction and spatial difference analysis method, which is beneficial to improve the quantitative degree of vulnerability spatial analysis. It can not only quantify regional drought vulnerability based on the disaster-causing mechanism but also convey vulnerability information to decision makers from a risk visualization perspective.

As wheat is one of the three major grain crops in the world, we select the main wheat producing area, the European winter wheat growing area, as the research area, using the $0.5° \times 0.5°$ grid as the basic assessment unit. The vulnerability curves of winter wheat drought are established based on EPIC simulation. Then, the loss extent and loss change characteristics of the vulnerability curve are extracted to analyse the vulnerability characteristics to drought in various areas. By clustering the curve shapes, areas with similar vulnerability characteristics are identified for exploring their environment and providing scientific guidance regarding the development of regional drought mitigation strategies.

## 2 Data and methods

### 2.1 Basic concept

Crop drought vulnerability curve describes the functional relationship between drought intensity and loss. As drought intensifies, disaster losses begin to appear and gradually increase until the end of the disaster. That is regarded as an interactive process of energy accumulation and resisting effect (Chen et al., 2015;Chen et al., 2017). Drought intensification brings about energy accumulation, which will be released when it reaches a certain level; meanwhile, resistance, such as system adjustment ability, always exists. In the initial stage, it appears as a slow development of drought due to insufficient energy storage

and the existence of resistance. And if the driving force is stopped or weakened, the energy accumulation basically ends; otherwise, energy will continue to accumulate, then break through the resistance and release, resulting in explosive development. Finally, the drought event gradually subsided with energy attenuation and resistance influence.

Therefore, the drought vulnerability curve can be divided into three stages as follows (Kucharavy and De Guio, 2011;Wang et al., 2013): (1) initial stage, corresponding to low drought intensity and slight loss, during which there is slow loss growth acceleration; (2) development stage, corresponding to moderate drought intensity and a rapid increase in loss, during which the loss growth rate continues to increase to reach a peak and then quickly falls; and (3) attenuation stage, corresponding to high drought intensity

and stable high loss, during which the loss growth rate slowly decays. These characteristics coincide well with the S-shaped curve (Fig. 1).

In different environments, the drought vulnerability curve presents different S-shapes (Wang et al., 2013;Yue et al., 2015;Guo et al., 2016), and the core lies in the differences in loss extent and loss change (Gottschalk and Dunn, 2005;Hu et al., 2012;Wang et al., 2013). Therefore, the key points of the

vulnerability curve—the transition points of three stages (P1 and P3, where the third derivative of the vulnerability curve is equal to zero) and the turning point of the loss growth rate (P2, where the second derivative of the vulnerability curve equals zero) are used to describe the loss change characteristics, the cumulative loss to the loss extent characteristics, and the morphological classification to of the integrated description.

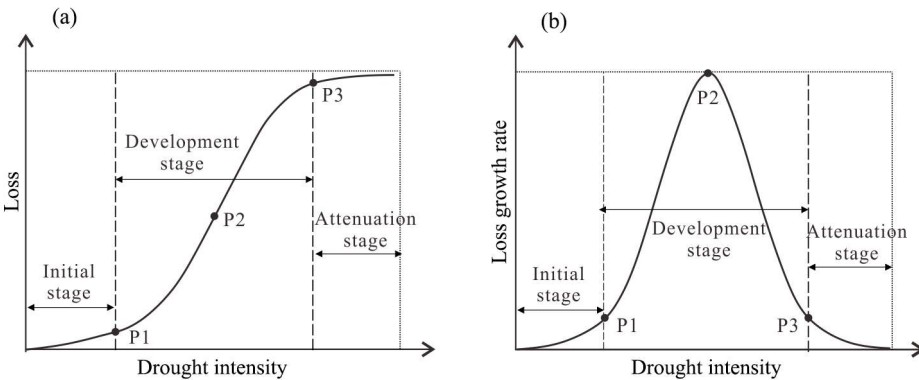

**Figure 1: The relationship between drought intensity and (a) loss and (b) loss growth rate as shown by the S-shape drought vulnerability curve. P1, P2, P3 represent the starting point, inflection point and end point of the rapid loss growth, respectively.**

**2.2 EPIC model and database construction**

The EPIC model, published by the United States in 1984 (Williams et al., 1984), is selected to simulate the growth process of winter wheat. It can simulate soil erosion and productivity for hundreds of years on a daily step under a variety of climatic, environmental and management conditions. It simulates all crops with one model framework based on crop's physiological commonality and uses unique crop parameters for each crop. In the process of simulation, intercepted photosynthetic active radiation is

converted into potential biomass, which is adjusted by five daily stress factors (water, nitrogen, phosphorus, temperature, and aeration) to predict actual biomass growth, where the water stress (WS) factor is computed as the ratio of soil water use over potential plant water use. Crop yields are estimated as the product of the actual above ground biomass and a harvest index (economic yield/above ground

biomass) (Williams et al., 1989).

EPIC model has been successfully applied in yield simulation for different crops and water input conditions in many parts of the world (Roloff et al., 1998;Gassman et al., 2005). Williams et al. (1989) described the EPIC model simulation results of 6 crop species throughout the U.S. and in European and Asian countries and concluded that the average simulated yields were always within 7 % of the average measured yields. Bryant et al. (1992) used the EPIC model to duplicate 38 irrigation stress experiments in the Texas High Plains during 1975-1977 and found that simulated corn yields explained 83, 86, and 72 % of the variance in 3-year measured yields separately. Ko et al. (2009) calibrated the EPIC model based on field studies in South Texas, and demonstrated that under full and deficient irrigation and rainfall conditions, EPIC-simulated yields of maize and cotton were in agreement with the measured yields according to a paired t-test.

With good performance in water stress tests, the model is further widely used in crop drought research, including irrigation management, drought impact prediction and drought vulnerability assessment (Guo et al., 2020). By setting different irrigation times, irrigation amounts and irrigation frequency to observe the EPIC-simulated yields, the optimized irrigation scheduling can be obtained without carrying out long and expensive field experiments (Rinaldi, 2001); by inputting climate model data, the future yield loss due to drought in different climate change scenarios can be prediction (Webber et al., 2018;Leng and Hall, 2019); and by using multi-year precipitation or irrigation data, a series of grid yield loss data can be obtained for quantifying drought vulnerability (Wang et al., 2013;Kamali et al., 2018c). In short, by setting up drought scenarios, the EPIC model can efficiently provide fine yield loss data. Therefore, we choose it as the core tool for drought vulnerability assessment.

The study area is the European wheat harvest area provided by the Center for Sustainability and the Global Environment, University of Wisconsin-Madison (Monfreda et al., 2008), and further screened by the wheat planting habit distribution map of CIMMYT (Lantican et al., 2005) for winter wheat distribution. Distributed in the range of 10° W-50° E and 42° N-59° N, this area is one of the world's major wheat-producing areas.

Inputs to EPIC include topography, soil, meteorological, and field management data (Table 1). The soil data in this study are provided by the International Soil Reference and Information Centre (Batjes, 2012), including soil type distribution raster maps and soil physical and chemical property lookup tables (soil bulk density, soil water content, grit content, clay content, organic carbon content, pH, etc.). The daily meteorological data are derived from HadGEM2-ES model data (Hempel et al., 2013) from 1974 to 2004, which are based on meteorological observations including solar radiation, maximum temperature, minimum temperature, average temperature, precipitation, relative humidity and average wind speed. All the original input data are processed onto $0.5° \times 0.5°$ grids, which are the basic units for the yield simulation and vulnerability assessment.

The statistical yield data are not required for EPIC model input but for the localization of crop parameters in the model and validation of simulated yields. They are derived from the Food and Agriculture Organization (FAO) and are country-based statistics. We use statistical yields of 2000 for model localization, and yields of other years between 1974 and 2004 for validation.

Outputs from the EPIC model include daily stress factors (water, nitrogen, phosphorus, temperature, and

aeration) and annual yield value. The WS and yield can be further processed into sample for the construction of vulnerability curves.

**Table 1: Basic database**

| Category | Name | Source | Spatial resolution |
|---|---|---|---|
| Distribution range data | Harvested area of wheat | Sustainability and the Global Environment, University of Wisconsin-Madison (Monfreda et al., 2008) | 5´×5´ |
| | Distribution of wheat planting habit | CIMMYT (Lantican et al., 2005) | Site unit |
| | Administrative boundary | Eurostat (https://ec.europa.eu/eurostat/web/gisco/ geodata/reference-data) | 1: 10 Million |
| Environmental data | DEM | United States Geological Survey (1996) | 0.5´×0.5´ |
| | Slope | Food and Agriculture Organization of the United Nations/International Institute for Applied Systems Analysis (http://www.iiasa.ac.at/Research/LUC/GAEZ/ index.htm, 2000) | 5´×5´ |
| | Soil | International Soil Reference and Information Centre (Batjes, 2012) | 5´×5´ |
| | Historical daily meteorological data (1974-2004) | German Federal Ministry of Education and Research: the ISIMIP Fast Track project (Hempel et al., 2013) | 0.5°×0.5° |
| Management data | Growth period of winter wheat | University of Wisconsin-Madison Sustainability and the Global Environment (Sacks et al., 2010) | 0.5°×0.5° |
| | Irrigation | OKI Laboratory, University of Tokyo (http://hydro.iis.u-tokyo.ac.jp/GW/result/ global/annual/withdrwith/index.html, 2002) | 0.5°×0.5° |
| | Fertilizer | Land Use and the Global Environment (Potter et al., 2010) | 0.5°×0.5° |
| Statistical yield data | Statistical yield for calibration (2000) | Food and Agriculture Organization of the United Nations (http://faostat.fao.org) | National (regional) unit |
| | Statistical yield for validation (1974-2004) | | |

5    **2.3 Research method**

This study consists of the following three parts. (1) Calibration and validation of the EPIC model. Critical crop parameters in the model are localized to improve the simulation accuracy in different regions. Then the calibrated model performance is validated by comparing simulated and statistical yields. (2) Construction of winter wheat drought vulnerability curves based on the calibrated EPIC model simulation.

A set of WS and yields are simulated for each grid unit by setting series of irrigation scenarios, which are converted into drought index and yield loss rate for the construction of the vulnerability curve. (3) Vulnerability curve characteristics analysis. Key points and cumulative loss rate of vulnerability curves are calculated for the spatial analysis of loss change and loss extent characteristics, and the vulnerability

curves are clustered for the integrated spatial analysis (Fig. 2).

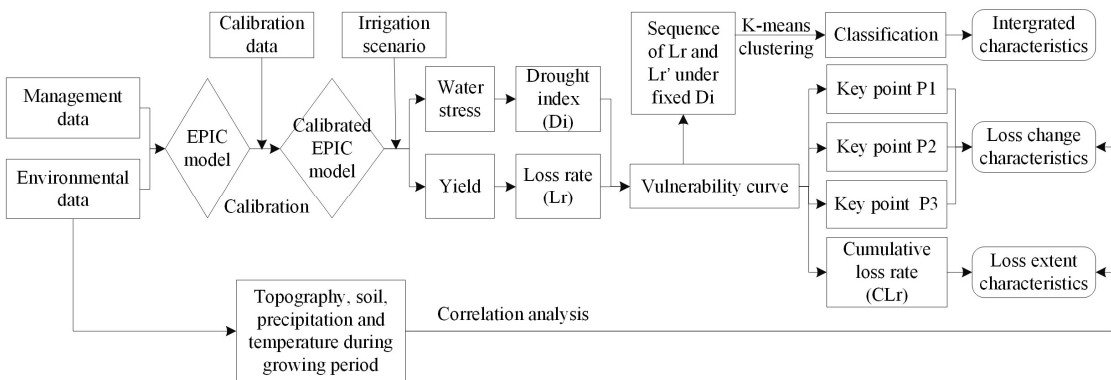

**Figure 2: Basic research framework. First, we input relevant data into the EPIC model and perform model calibration. Next, we obtain a series of water stress and yield data based on the calibrated EPIC model by**

**setting different irrigation scenarios, which are converted into drought index (Di) and yield loss rate (Lr) for the construction of vulnerability curves. Then, we extract three key points and calculate the cumulative loss rate of vulnerability curves for the spatial analysis of loss change and loss extent characteristics. Finally, we calculate the Lr and the growth rate of Lr (Lr') under a set of fixed Di to transform the vulnerability curves into a finite data set for clustering, and the classification of vulnerability curves can be used for the integrated**

**spatial analysis.**

### 2.3.1 Calibration and validation of the EPIC model

The calibration method refers to the research of Guo et al. (2016). Four key parameters of WA (biomass-energy ratio), HI (harvest index), DLMA (maximum potential leaf area index), and DLAI (fraction of the growing season when the leaf area decreases) are selected for calibration (Barros et al., 2005; Wang and

Li, 2010; Wang et al., 2011). Considering the limitation of statistical yields on a grid scale, we localize the four key parameters at the country level based on the idea of partition calibration (Liu et al., 2007; Balkovič et al., 2013; Kamali et al., 2018a). That is, each country has a unique set of crop parameters, and all the grids within one country are the same. The default values of the crop parameters in the EPIC model are taken as the initial value, and the geographical environmental, field management and

meteorological data are entered to obtain simulated grid yields of 2000. We simply assign a FAO national statistical yield to the grids within a country. Then the root mean square error (RMSE) between the simulated and statistical grid yields for each country are calculated. We reiterate the yield simulations and RMSE calculations by incrementally adjusting the four key parameters to minimize RMSE. The calibration will be finished when the least RMSE is below the threshold or the number of reiterations is

above the threshold.

To validate the parameterization results, we generate the simulated grid yields of 1974-2004 based on the calibrated EPIC model, and aggregate to the nation level by averaging. For FAO national statistical yields of 1974-2004 with significant trends, linear de-trending transformations are applied to remove the impacts of technology progress (Xiong et al., 2014; Kamali et al., 2018a). Then we compare national

simulated yields with the statistical yields across all European countries.

**2.3.2 Vulnerability curve construction based on the calibrated EPIC model**

**(1) Generation of WS and yields under different irrigation scenarios**

After parameter localization, the EPIC model can be used to simulate WS and the winter wheat yields under different drought scenarios, providing samples for the construction of vulnerability curves to drought.

To focus on physical drought vulnerability and eliminate the impact of other stress factors on yields, we use meteorological data with suitable temperature and no precipitation, and control the water supply condition by setting 20 irrigation scenarios, in which the irrigation amount uniformly increases from 0 to the optimum (the maximum irrigation amount without WS). The optimal value is determined by pre-testing. Consequently, we obtain the outputs of 20 groups of WS and yield for each grid evaluation unit.

**(2) Calculation of the drought index and yield loss rate**

As an output factor of the EPIC model, WS reflects the relationship between daily water supply and crop water demand. WS ranges from 0-1; the larger the value, the more serious the water shortage will be. To characterize integrated drought intensity affecting yield, drought index (Di) is defined as relative cumulative water stress during the crop growth period, which can reflect both WS intensity and stress duration (Wang et al., 2013). The calculation is shown in Eq. (1) and Eq. (2):

$$Di_i = \frac{HI_i}{\max(HI)} \ , \tag{1}$$

$$HI_i = \sum_{d=1}^{n}(WS_k) \ , \tag{2}$$

where $Di_i$ is the drought index of a grid unit under the irrigation scenario $i$, ranging from 0-1; $HI_i$ is the cumulative value of $WS$ during the growth period under this scenario; $\max(HI)$ is the maximum value of $HI_i$ under all irrigation scenarios; $WS_k$ is the $WS$ value on day k of the growth period; and n is the number of days affected by $WS$ during the growth period.

The yield loss rate (Lr) is used to express the response of the yield to drought effects, calculated following Eq. (3):

$$Lr_i = \frac{\max(y) - y_i}{\max(y)} \ , \tag{3}$$

where $Lr_i$ is the yield loss rate of a grid unit under irrigation scenario $i$, $y_i$ is the yield under this scenario and $\max(y)$ is the maximum yield under the optimal irrigation scenario.

**(3) Fitting of drought vulnerability curves**

The aforementioned Di-Lr samples are fitted by a logistical curve to obtain the grid vulnerability curve, as shown in Eq. (4):

$$Lr = \frac{a}{1 + b \times e^{c \times Di}} + d \ , \tag{4}$$

where a, b, c, and d are constant parameters.

Then the coefficient of determination ($R^2$) and RMSE are used to measure the imitative effect (Quiring and Papakryiakou, 2003). $R^2$ represents the proportion of the total variance in the observed Di-Lr samples that can be explained by the fitting model. It ranges from 0 to 1, where the higher values indicate better fitting accuracy. RMSE represents the average difference between the predicted values by the fitting model and the observed samples, and the higher values indicate worse fitting accuracy.

### 2.3.3 Feature extraction and spatial analysis of the vulnerability curves

**(1) Identification of key points**

According to the analysis in Section 2.1, taking the derivative of Eq. (4), and setting the second and third derivatives equal to 0, the coordinates of the key points can be obtained to characterize the phase change in the vulnerability curve (Table 2).

**Table 2: Key point coordinates of the vulnerability curve**

|  | The starting point of rapid loss growth (P1) | The inflection point of rapid loss growth (P2) | The end point of rapid loss growth (P3) |
|---|---|---|---|
| Di | $-\frac{\ln(2-\sqrt{3})b}{c}$ | $-\frac{\ln b}{c}$ | $-\frac{\ln(2+\sqrt{3})b}{c}$ |
| Lr | $\frac{(3-\sqrt{3})a}{6}+d$ | $\frac{a}{2}+d$ | $\frac{(3+\sqrt{3})a}{6}+d$ |

**(2) Calculation of the cumulative loss rate**

The cumulative loss rate (CLr) is obtained by the integral of Eq. (4) on the Di interval of [0,1] to describe the overall vulnerability. All CLr values are divided into five levels by the natural breakpoint method: extremely low (0.22-0.34), low (0.34-0.42), moderate (0.42-0.49), high (0.49-0.55), and extremely high (0.55-0.69).

**(3) Clustering of the vulnerability curves**

To identify the morphological characteristics of the vulnerability curves, the curves are divided into some categories by clustering. The first step is to filter the infinite dimensional curve data to a finite set of representative parameters (James and Sugar, 2003). A set of Lr and growth rate of Lr (Lr') under the fixed Di (0.2, 0.4, 0.6, and 0.8) are selected to preserve both the loss extent and change characteristics of a curve, where distinguishing the differences between the curves. The 8 elements are separately normalised following Eq. (5) for clustering.

$$N(Lr_{Di=x})_t = \frac{(Lr_{Di=x})_t}{\text{SD}(Lr_{Di=x})} , \tag{5}$$

where $(Lr_{Di=x})_t$ is the value of Lr (Lr') when Di=x for the vulnerability curve t, and x=0.2, 0.4, 0.6, and 0.8; $\text{SD}(Lr_{Di=x})$ is the standard deviation of Lr (Lr') when Di=x for all vulnerability curves; and $N(Lr_{Di=x})_t$ is the normalised value.

The second step is to choose an appropriate clustering tool. Generally, clustering algorithms can be divided into four categories, partitional clustering, hierarchical clustering, grid-based clustering and density-based clustering. Partitional clustering directly divides the data set into several sub-sets without intersection; hierarchical clustering creates a hierarchical decomposition of the data set to perform clustering, and it cannot be traced back after classifying; density-based clustering controls the growth of clusters through judging the relationship of data density (the number of instances in unite area) and threshold; grid-based clustering divides the object space into a limited number of cells to form a grid structure, and is often combined with other methods, especially density-based clustering methods (Sun et al., 2008;Han et al., 2012).

K-means is a clustering algorithm which based on partition. It has the characteristics of faster calculation speed and good clustering effect, which has been widely used in clustering research (Sun et al., 2008;Wu

et al., 2011). This paper uses the basic K-means clustering method to the representative parameter set, and uses Euclidean distance to compare the similarity of vulnerability curves among grid cells (Jacques and Preda, 2014). The smaller the distance, the more similar of the vulnerability curves. The steps of the K-means clustering are:

① Setting up the classification number K of the data set, and then randomly select K data points from the data set as initial centroid of the classification;

② Sorting each data point into the class which are closest it;

③ Calculating the new centroid of each class based on all data points in it;

④ Repeating steps ② and ③ until the centroid of each class remains the same or reaches the limit of iterations.

The selection of K value is the key of K-means clustering. The elbow method is a commonly used method for selecting K values, which is based on the sum of squares of errors (SSE) (Nainggolan et al., 2019;Wang et al., 2019). The SSE is calculated as follows:

$$SSE = \sum_{i=1}^{K} SSE_i \qquad (6)$$

$$SSE_i = \sum_{i=1}^{n} (P_{i,j} - C_i)^2 \qquad (7)$$

Where $SSE_i$ is the sum of squares of errors within the i-th class, $n$ is the number of data points in the i-th class, $C_i$ is the centroid of the i-th class, $P_{i,j}$ is the j-th sample point in the i-th class.

The principle of the elbow method is as follow (Nainggolan et al., 2019;Wang et al., 2019). As the K value increases, the data set becomes finer, the data points are closer to the centroid, and the SSE will become smaller. The increase of the K value will reduce the SSE greatly and improve the clustering effect in the early stage; but when it reaches a certain value, the SSE will decline slowly and over-classification may occur. That is to say, the K value has an elbow relationship with the SSE, and the elbow point corresponds to optimal number of clusters. Considering that the elbow point may not be obvious, we have further counted the number of vulnerability curves in each cluster with different K values, so as to determine the optimal K value comprehensively.

After clustering, the further category vulnerability curves are fitted by all the Di-Lr samples in corresponding cluster, for better describing the characteristics of each cluster.

## 3 Results and analysis

### 3.1 Validation of the EPIC model simulation results

From the national comparison results from 1974 to 2004 (excluding calibration year of 2000), though the simulated yields are slightly higher than the statistical yields, there is high agreement between the two (Fig. 3). The regression equation has an $R^2$ of 0.77 and passes the test with a confidence of 0.01, indicating a reliable performance of the calibrated EPIC model for yields simulation in various regions and various years.

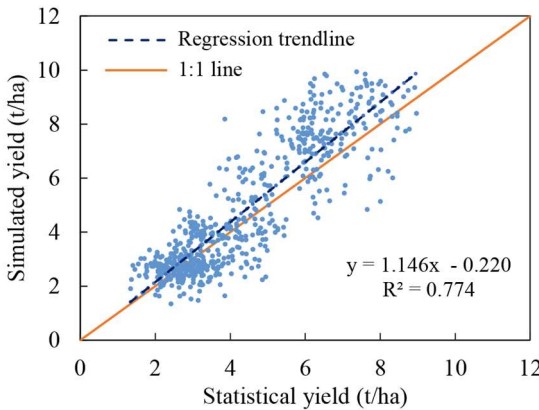

**Figure 3: Comparison of national winter wheat yield reported by FAO and simulated by calibrated EPIC during the period from 1974-2004 (excluding calibration year of 2000).**

### 3.2 European winter wheat drought vulnerability curves and characteristics analysis

5 ### 3.2.1 Winter wheat drought vulnerability curves

Figure 5a shows the drought vulnerability curves of the 2010 grid assessment unit in Europe. On the grid scale, the $R^2$ values of the vulnerability curve fitting are all above 0.94, and 97.5 % of them are above 0.996 (Fig. 4a). Grids with $R^2$ less than 0.999 are mainly distributed in Ukraine, Germany, Macedonia, Greece. The RMSE values are concentrated between 0-0.043, and the 94.5 % of them are less than 0.02

10 (Fig. 4b). Grids with RMSE values greater than 0.15 are mainly belong to Ukraine. In general, the $R^2$ of the regional vulnerability curve fitted by all the grid Di-Lr samples is 0.90 and the RMSE is 0.12, indicating a high overall goodness of fit.

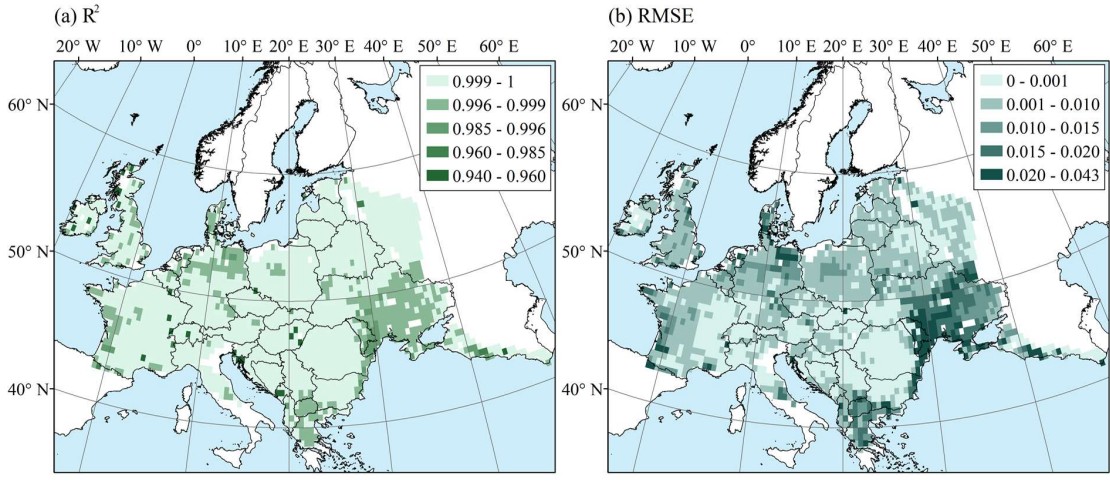

**Figure 4: Goodness of fit of grid vulnerability curves, including $R^2$ and RMSE measures.**

There are differences in the shape of vulnerability curves and in the coordinates of key points (Fig. 5b). The regional starting point, inflection point and end point of the rapid loss of growth correspond to Di values of 0.27, 0.47 and 0.68 and Lr values of 0.17, 0.43 and 0.75, respectively. For most grids, the Di values at the three key points are mainly distributed from 0.15-0.55, 0.35-0.7 and 0.4-0.8, while the Lr

20 values have a relatively small distribution, from 0.1-0.2, 0.4-0.5 and 0.7-0.8. Therefore, the characteristics of stage transitions of grid vulnerability curves can be simplified by using the Di instead

of two coordinates at key points. The larger the Di is at key points, the more severe the drought must be to cause a similar loss rate; this is reflected in the lag in the stage transitions of vulnerability curve, indicating a greater tolerance to drought disturbance.

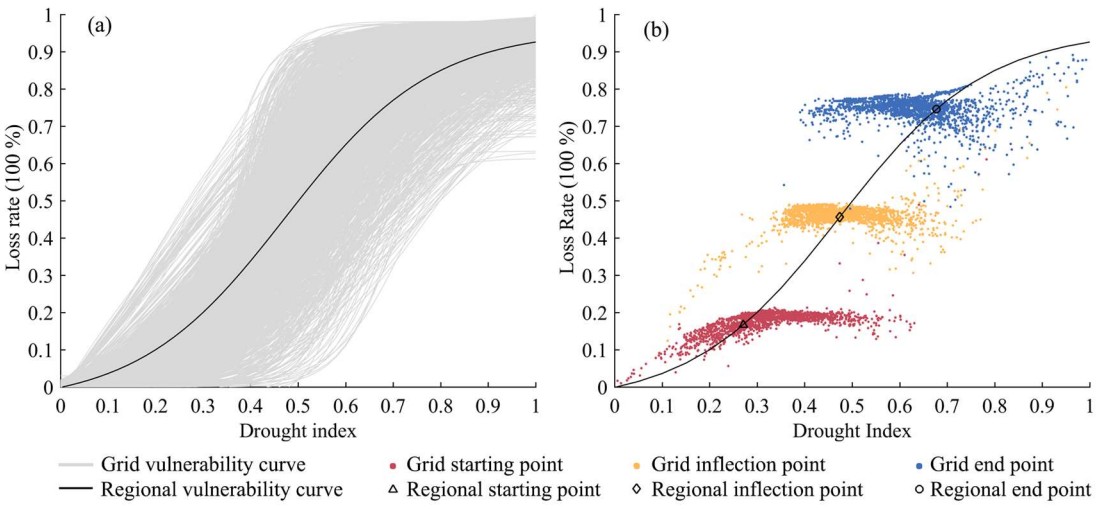

Figure 5: Distribution of (a) regional and grid vulnerability curves and (b) their three key points. The regional vulnerability curve is fitted by all drought index-loss rate sample data in the region.

### 3.2.2 Spatial distribution of the characteristic value

In terms of spatial distribution, the Di values at key points in the south are higher than those in the north (Fig. 6). In the southern areas, the Di values at the starting, inflection and end points are concentrated in 0.4-0.5, 0.5-0.7, and greater than 0.7, respectively, while in north-central areas, they are less than 0.2, 0.3-0.5, and 0.5-0.7, respectively. Therefore, the stage transitions of the vulnerability curves in the southern areas lag behind, indicating a higher tolerance to drought disturbance. In the northeast, the Di values at the start and end points are within the range of 0.2-0.4 and 0.4-0.6, respectively, indicating that Lr changes drastically during a short development stage, during which these areas are particularly susceptible to drought.

The CLr represents the overall vulnerability, which is contrary to the meaning of Di at key points, and naturally shows an opposite distribution of low in the south and high in the north. Though both the north-central areas and the northeast areas have extremely high CLr values, stage transition characteristics in the two areas are different. The CLr integrates the characteristics of the key points but shows information loss in the characteristics of loss change.

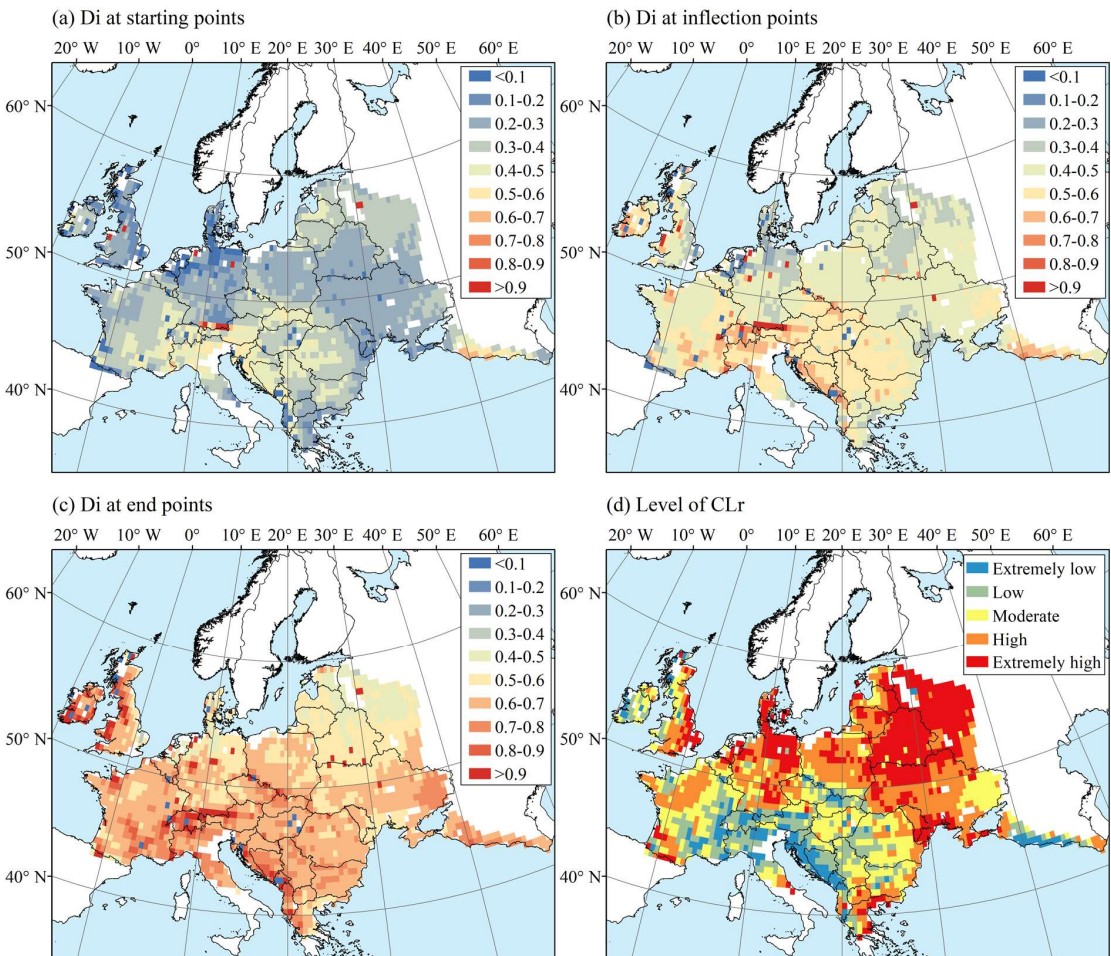

**Figure 6: Spatial distributions of drought index (Di) at the (a) starting points, (b) inflection points and (c) end points, and (d) spatial distribution of the level of the cumulative loss rate (CLr) of vulnerability curves.**

### 3.3 Categories of winter wheat drought vulnerability curves

To comprehensively analyse the vulnerability types of regions, we convert the vulnerability curve into a representative parameter set of loss degree and loss change characteristics (Appendix A), and then perform K-means clustering. When determining the optimal number of classification (K value), it is found that when K=5, the line graph of SSE shows an inflection point (Fig.7); at this time, the number of instance in each cluster is relatively uniform, so as not to be over-concentration or over-classification (Table 3), indicating an optimal classification effect. Therefore, the grid drought vulnerability curves are divided into 5 categories.

Compared to the regional loss characteristics at the initial, development and attenuation stages, these types of vulnerability curves are defined as Low-Low-Low (L-L-L), Low-Low-Medium (L-L-M), Medium-Medium-Medium (M-M-M), High-High-High (H-H-H) and Low-Medium-High (L-M-H) loss-type vulnerability curves (Fig. 8). Five category vulnerability curves are fitted based on the Di-Lr samples of related vulnerability curves for a comprehensive characterization.

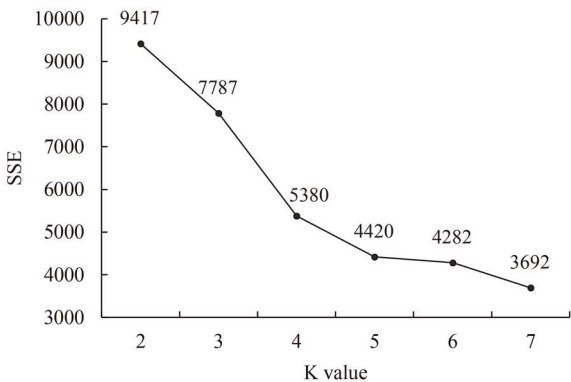

**Figure 7: Sum of squared errors (SSE) corresponding to different clustering numbers (K).**

**Table 3: Clustering effect of different cluster quantities.**

| Quantity of cluster (K) | Quantity of vulnerability curves in each cluster | | | | | | |
|---|---|---|---|---|---|---|---|
| | Cluster 1 | Cluster 2 | Cluster 3 | Cluster 4 | Cluster 5 | Cluster 6 | Cluster 7 |
| 2 | 739 | 1271 | - | - | - | - | - |
| 3 | 232 | 1076 | 702 | - | - | - | - |
| 4 | 254 | 637 | 886 | 233 | - | - | - |
| 5 | 472 | 407 | 686 | 243 | 202 | - | - |
| 6 | 3 | 471 | 683 | 407 | 243 | 203 | - |
| 7 | 3 | 155 | 706 | 156 | 193 | 254 | 543 |

The Lr values of the L-L-L loss-type vulnerability curves are lower than the regional level under the same Di, and the category CLr is only 0.33 (calculated by the category vulnerability curve), which is the lowest value of the five categories (Appendix B). These vulnerability curves are mainly distributed in mountain areas such as the Alps and the Dinara and Caucasus mountains, accounting for 10.0 % of the winter wheat planting area in Europe.

The L-L-M loss-type vulnerability curves have a relatively low loss rate and are susceptible to drought within the range of 0.4-0.7. When the Di values reach approximately 0.4, the loss rates begin to rapidly increase; when the Di values are greater than 0.6-0.7, the loss rates are near the regional level. The category CLr is 0.42. It is mainly found in the Danube river basins, including hilly areas and plains, accounting for 20.3 % of the winter wheat planting area in Europe.

The M-M-M loss-type vulnerability curves are near the regional vulnerability curve with a category CLr of 0.50, and mainly occur in the Western European Plains, the Pod Plains, Donets Ridge and surrounding highlands and lowlands. They have the widest distribution accounting for 34.1 % of the winter wheat planting area in Europe.

The Lr values of the H-H-H loss-type vulnerability curves are higher than the regional level, and the category CLr reaches 0.57. These vulnerability curves are concentrated in patches on the Pod Plain, Polesi and in lowland areas along the Black Sea and Eastern Great Britain, at approximately the same latitude zone as that of the M-M-M loss-type, accounting for 23.5 % of the winter wheat planting area in

Europe.

The L-M-H loss-type vulnerability curves show high susceptibility to drought in the range of 0.3-0.6, where the Lr values rapidly increase and reach the regional level with the increase in Di. When Di values are greater than 0.6 and continue to increase, the Lr values maintain relatively stable and high level; when Di values are less than 0.3, the Lr values are slight. The category CLr is 0.53. These curves are mainly distributed on the east European plain, accounting for 12.1 % of the winter wheat planting area in Europe. Overall, the spatial distributions of the five types of vulnerability curves are obviously latitudinal and consistent with the geographical pattern of Europe, where plains and mountains mostly extend from the east to the west in the mainland and extend from north to south in the British Isles. From south to north, and from mountain to plain, the vulnerability curves transition from concave to convex, and the CLr values show an upward trend, indicating increasing vulnerability. The heat difference at different latitudes and the water and heat difference at different altitudes may be the root cause of the type distribution.

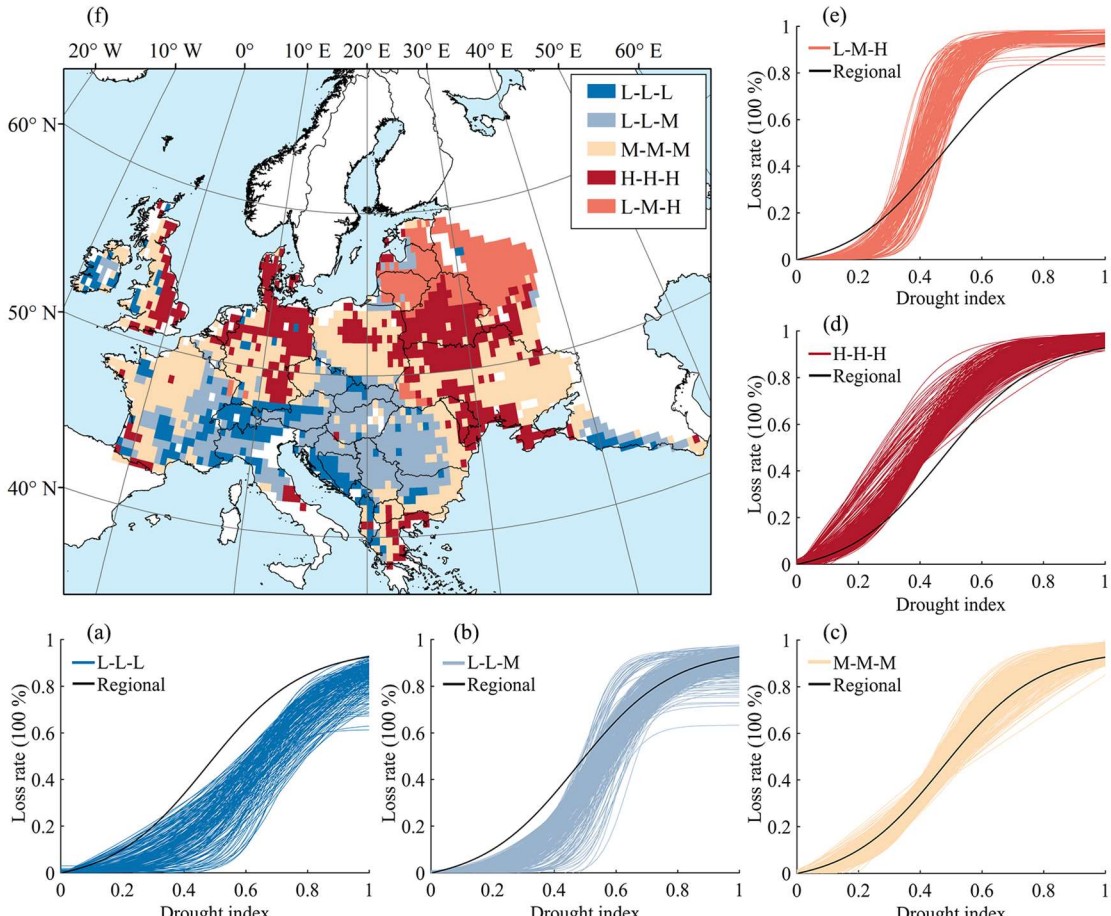

**Figure 8: Five types of European winter wheat vulnerability curves to drought: (a) L-L-L, (b) L-L-M, (c) M-M-M, (d) H-H-H and (e) L-M-H loss-type vulnerability curves, and (f) their spatial distributions.**

## 4 Discussion

### 4.1 Relationship between vulnerability characteristics and environmental variables

To further explore the relationship between the vulnerability characteristics distribution and environmental variables, Spearman correlation analysis is performed between the vulnerability

characteristics parameters ($Di_1$, $Di_2$, $Di_3$, and CLr) and environmental variables (elevation, slope, soil sand content, precipitation during growth period, average temperature during growth period, and relative humidity during growth period). The results all passed the significance test at the level of 0.01 (Table 4). The $Di_1$ value is positively correlated with relative humidity and elevation, and the correlation coefficients are 0.41 and 0.40, respectively. That is, in areas with high relative humidity or altitude, only when the drought develops to a rather serious extent does it begin to have a significant impact on winter wheat yield. Additionally, the L-L-L, L-L-M and L-M-H loss-type areas with high $Di_1$ values have the characteristics of high elevation or high relative humidity (Appendix C).

The four characteristic parameters are highly correlated with the environmental variables with latitudinal zonality, such as elevation, slope, temperature and soil sand content, which verifies the inference of the distribution of the characteristic parameters above. The $Di_1$, $Di_2$ and $Di_3$ values characterizing drought tolerance are positively correlated with elevation, slope and temperature, and negatively correlated with soil sandy content, while the CLr value characterizing the comprehensive vulnerability shows the opposite trend. The H-H-H loss-type areas with high vulnerability have typical characteristics of low elevation, slope, temperature and high soil sandy content.

From the perspective of an influencing mechanism, when the soil sandy content is high, the soil drainage ability is strong, and the crop is more vulnerable to drought (Reid et al., 2006;Papathoma-Köhle, 2016) , exhibiting low $Di_1$, $Di_2$, and $Di_3$ values and a high CLr value in the vulnerability curve. The cause-effect relationship between the temperature and the characteristic parameters cannot be defined, although the spatial distributions of the two have a certain correlation. Because temperature stress is removed from the drought scenarios, the temperature variable has no direct influence on the results of yield loss rate to drought and the characteristic parameters. It may have an indirect influence by affecting the crop parameters of winter wheat during the previous calibration process. Similarly, elevation does not directly affect the values of the characteristic parameters. Simulation experiments based on the EPIC model found that changing the input of elevation has little effect on the simulated yield (Thomson et al., 2002). Thus, the elevation may indirectly affect yield and drought vulnerability by acting on other environmental variables such as temperature, precipitation and soil. The aforementioned can provide ideas for studying the impact of the environment on vulnerability.

**Table 4: Correlation between vulnerability characteristic parameters and environmental variables (P≤0.01)**

| Environmental variable | $Di_1$ | $Di_2$ | $Di_3$ | CLr |
|---|---|---|---|---|
| Elevation | 0.40 | 0.43 | 0.37 | -0.44 |
| Slope | 0.31 | 0.44 | 0.45 | -0.48 |
| Soil sand content | -0.10 | -0.35 | -0.44 | 0.38 |
| Average temperature during growth period | 0.32 | 0.34 | 0.30 | -0.38 |
| Precipitation during growth period | -0.09 | 0.19 | 0.33 | -0.26 |
| Relative humidity during growth period | 0.41 | 0.23 | 0.09 | -0.27 |

**4.2 Uncertainty and limitation**

The EPIC model default crop parameters may deviate from the actual growth in different regions, so we localize and verify the crop parameters to be as close to reality as possible. Nevertheless, there are some

inevitable uncertainties, derived from the selection of calibrated crop parameters, the accuracy of the statistical yield data, and other factors. There are 56 crop parameters in the EPIC model, and different input parameters have different degrees of influence on the EPIC model in different simulation environments (Zhang et al., 2017). The main method to reduce the uncertainties of input parameters is to

carry out sensitivity analysis in the basic evaluation unit and calibrate the sensitivity parameters one by one. However, this requires multiple calculations and does not completely eliminate the uncertainties of the input parameters (Yue et al., 2018). Therefore, with reference to previous research, we focus on the calibration and validation of the above four main sensitive parameters. In terms of the accuracy of the statistical yield, we use national-scale data due to the availability, which is coarser than the grid

simulation unit, so it may cause some uncertainties in the localization and verification results. When more multi-year and higher-resolution statistical yield data are available in the future, the results will be further improved.

There may also be uncertainties in the process of vulnerability simulation and assessment using the calibrated EPIC model. To quantify them, we reiterate this process 20 times and evaluate the standard

deviation distribution of the results. First, we randomly select 10 % of samples from the five types of vulnerability curves based on the principle of stratified sampling, and obtain a total of 201 sample grids. Next, according to the method in Section 2.3.1, we reiterate the vulnerability simulation and vulnerability curve construction process 20 times by changing the irrigation scenario settings, that is, keeping the non-irrigation and optimal irrigation scenarios unchanged and then randomly setting 18 irrigation scenarios

between the two. From this, 20 reiterated vulnerability curves can be obtained for each sample grid. Then, by calculating the standard deviation of the Lr for 20 reiterated vulnerability curves at the drought index interval of 0.1, the standard deviation of Lr for each sample grid can be obtained to characterize the grid uncertainties. The mean standard deviation and 95 % prediction uncertainty band (95PPU) of total sample grids are finally calculated to characterize overall uncertainties. 95PPU is the range from 2.5 % to 97.5 %

of the cumulative distribution function (Abbaspour et al., 2007). The results show that the mean standard deviation of Lr is between 0 and 0.065, and the average is 0.033; the width of PPU95 is between 0.007 and 0.135, and the average is 0.067; the two indicators reach the peak when the drought index is between 0.4 and 0.7 (Fig. 9). Although the prediction uncertainty of Lr is relatively large in such range, it is still significantly smaller than the differences in Lr between regions (which can reach more than 0.5), so it

has little effect on the distribution pattern of vulnerability. In summary, the uncertainties in this process are acceptable.

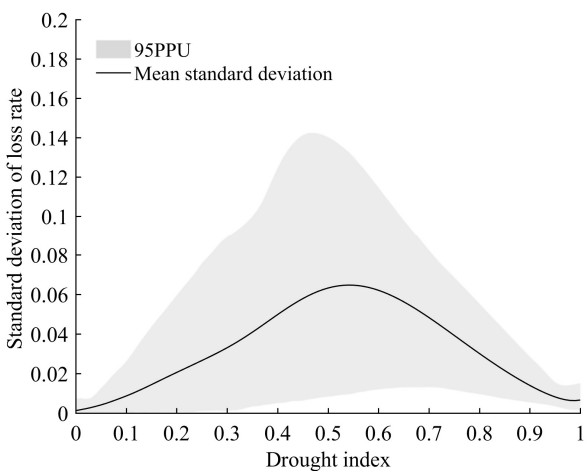

**Figure 9: Distribution of standard deviation of loss rate under different drought index. The mean standard deviation and 95 % prediction uncertainty band (95PPU) are calculated by the standard deviations of sample grids, which are randomly selected from the five vulnerability curves at a proportion of 10 %.**

**4.3 Prospection of the vulnerability curves**

By analysing the distribution of characteristic parameters, it is found that the winter wheat vulnerability in Europe is lower to the south, particularly in the surrounding areas of the Mediterranean, which is consistent with research findings based on experimental results of wheat varieties (Mäkinen et al., 2018) and the crop model simulation results at country scale (Leng and Hall, 2019).

By reflecting the spatial differences in vulnerability, the characteristic information can accurately express the response feature to drought in various regions and more effectively guide drought risk management. We suggest paying more attention to moderate and severe drought mitigation in southern Europe (mainly the L-L-L and L-L-M loss-type areas), improving the prevetion and mitigation capacity in the central region (mainly M-M-M and H-H-H loss-type areas), and seizing the susceptibility stage of drought development for mitigation in the north-eastern region (the L-M-H loss-type areas).

In addition, the vulnerability curve based on the crop growth process simulation helps to understand the risk from a vulnerability perspective. The impact of climate change on crop yield depends not only on the temporal and spatial patterns of climate change but also on species characteristics (Trnka et al., 2014;Semenov et al., 2014). From the perspective of climate change, the drought risk in southern Europe is more likely to increase compared to other regions of Europe, due to the predicted reduced precipitation and increased evaporation (Olesen et al., 2011;IPCC, 2012). However, it was found that the increase in drought effects on wheat in the southern region may be less than or near those of the central and north-eastern regions (Webber et al., 2018), which may be related to a lower drought vulnerability. This is also an indirect verification of the spatial difference analysis results in this paper.

Conducting a comprehensive vulnerability assessment combined with social vulnerability will be an important direction for future research. The vulnerability assessment will focus on the agricultural social ecosystem rather than crops. On the basis of consideration of variety characteristics and natural environmental factors, the impact of field farming measures such as regional irrigation, fertilization, and pest management should also be considered (González Tánago et al., 2016;Guo et al., 2020). In the further research, we suggest adding socio-economic factors into the crop growth simulation as field management parameters, such as irrigation capacity and fertilization level. It will improve the level of

evaluation and application value of regional vulnerability.

On the other hand, how to carry out dynamic vulnerability assessment needs further exploration. With climate change and socio-economic development, the crop planting dates, growth periods, irrigation and fertilization management may change constantly (Moriondo et al., 2010). The future vulnerability curves may be different from the current ones here. Therefore, it is recommended to explore dynamic vulnerability assessment methods, combining possible scenarios of climate change and socio-economic development, and then evaluate differences the comprehensive drought vulnerability under different scenarios. This work has important reference value for dynamic risk assessment and risk management.

**5 Conclusion**

Quantitative crop-drought vulnerability assessment and analysis are an important basis for drought risk assessment and drought risk management. Taking European winter wheat as an example, we generate series data of WS and scenario yield based on EPIC model simulation and then construct S-type drought vulnerability curves. Through characteristic parameters analysis and clustering analysis of vulnerability curves, the loss extent and loss change characteristics are mapped to identify the regional vulnerability pattern and drought response characteristics. The results provide quantitative ideas for the study of the impact of the environment on vulnerability and provide scientific guidance for regional drought mitigation resource allocation and strategy development.

The winter wheat drought vulnerability in Europe is higher in the south and lower in the north with a latitudinal zonality, which may be related to environmental variables such as elevation, slope, average temperature during growth period and soil sand content. In the southern region, the Di values at the key points are high, and the CLr values are low, indicating a low vulnerability, while the northern region shows the opposite trend.

The vulnerability curves can be divided into five loss types: L-L-L, L-L-M, M-M-M, H-H-H and L-M-H. It is recommended to improve the ability to address drought with a greater than 0.4 intensity in the L-L-L or L-L-M loss-type areas and a drought range from 0.3-0.6 intensity in the L-M-H loss-type areas, as well as improve drought prevention and mitigation in the M-M-M or H-H-H loss-type areas.

**Appendices**

**Appendix A: Spatial distribution of yield loss rate and loss rate growth rate under different drought index**

## Yield loss rate (Lr)          ## Growth rate of yield loss rate (Lr')

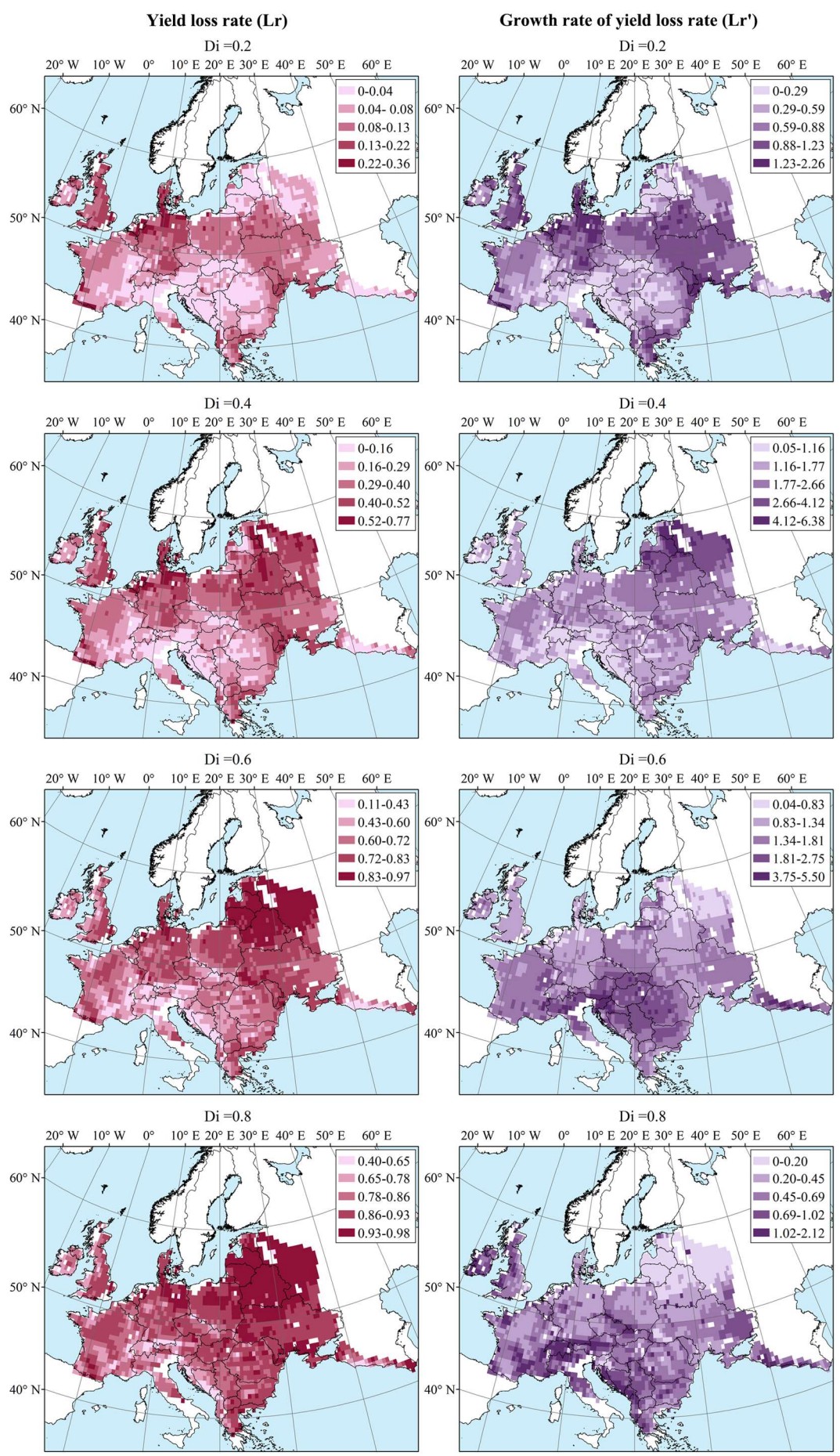

**Appendix B: Classificatory key points and cumulative loss rates calculated by category vulnerability curves**

| Category vulnerability curve | $Di_1$ | $Lr_1$ | $Di_2$ | $Lr_2$ | $Di_3$ | $Lr_3$ | CLr |
|---|---|---|---|---|---|---|---|
| L-L-L | 0.44 | 0.19 | 0.67 | 0.48 | 0.90 | 0.76 | 0.33 |
| L-L-M | 0.40 | 0.19 | 0.55 | 0.46 | 0.69 | 0.73 | 0.42 |
| M-M-M | 0.28 | 0.18 | 0.47 | 0.47 | 0.65 | 0.75 | 0.50 |
| H-H-H | 0.19 | 0.15 | 0.38 | 0.45 | 0.57 | 0.76 | 0.57 |
| L-M-H | 0.33 | 0.19 | 0.44 | 0.47 | 0.56 | 0.75 | 0.53 |
| Europe | 0.27 | 0.17 | 0.47 | 0.46 | 0.68 | 0.75 | 0.48 |

**Appendix C: Descriptive statistics of environmental variables in various loss-type regions**

| | | L-L-L | L-L-M | M-M-M | H-H-H | L-M-H | Regional |
|---|---|---|---|---|---|---|---|
| Elevation (m) | Median | 677 | 315 | 165 | 140 | 160 | 181 |
| | Interquartile Range | 636 | 468 | 154 | 125 | 103 | 241 |
| Slope (°) | Median | 23 | 12 | 6 | 3 | 3 | 6 |
| | Interquartile Range | 25 | 17 | 9 | 3 | 3 | 9 |
| Soil sand content (%) | Median | 43 | 43 | 43 | 52 | 52 | 43 |
| | Interquartile Range | 4 | 10 | 22 | 9 | 0 | 12 |
| Precipitation during growth period (mm) | Median | 960 | 646 | 599 | 599 | 638 | 629 |
| | Interquartile Range | 306 | 198 | 128 | 131 | 53 | 158 |
| Average temperature during growth period (℃) | Median | 7.1 | 7.8 | 7.5 | 6.9 | 3.9 | 7.1 |
| | Interquartile Range | 3.5 | 3.6 | 2.1 | 1.9 | 1.1 | 2.9 |
| Relative humidity during growth period (%) | Median | 79.9 | 80.6 | 77.5 | 77.1 | 80.2 | 78.8 |
| | Interquartile Range | 2.7 | 3 | 3.9 | 3.1 | 2.1 | 3.9 |

5    **Data availability**

The sources of raw data can be found in section 2.2. The code is written for MATLAB, which is available upon request by contacting Yanshen Wu (wuyanshen1012@mail.bnu.edu.cn).

**Author contribution**

Jing'ai Wang proposed overarching idea and formulated overarching research goals and aims. Hao Guo
10   implemented EPIC model calibration, simulation and vulnerability curves construction. Yanshen Wu and Anyu Zhang developed the vulnerability curves characteristics analysis methods and carried them out.

Yanshen Wu drafted and revised manuscript with contributions from all co-authors.

**Competing interests**

The authors declare that they have no conflict of interest.

**Acknowledgements**

The National Key Research and Development Program (No. 2016YFA0602402) and the National Basic Research Program of China (No. 2012CB955403) financially supported this research.

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
