# Peer review of "Establishment and characteristics analysis of a crop-drought vulnerability curve: a case study of European winter wheat"

_Natural Hazards and Earth System Sciences, 2019_

## Referee Comment (RC1) · Anonymous Referee #1 · 23 Sep 2019

This paper uses the drought vulnerability curves with three key points to assess wheat drought vulnerability in European area. The vulnerability curves are divided into five clusters and are compared with environmental variables. The utilization of key ponits and clustering classification of vulnerability curves is a good way to analyze crop drought vulnerability and its spatial distribution. However, the English writing of the manuscript should be improved. Some sentences are hard to understand. In addition, the definition of drought index is not clear. More explanation should be added. Therefore, the following questions should be modified before publication. There is no systematic introduction to the basic information of the Erosion-productivity Impact Calculator (EPIC) model or the research progress of this model in crop drought field. The

exposition of the theory and practice progress regarding crop drought vulnerability is insufficient in this research. It needs to be supplemented with a large amount of literature, especially those in the past 15 years, to discuss the historical research process of vulnerability assessment, which is from the single index to the linear index, and then to the curve index. The time series of meteorological data is too short to prove the credibility of the vulnerability simulation results. The spatial resolution of crop yield data is inconsistent with the spatial resolution of the simulation evaluation unit. It brings uncertainties to the verification of the model and the localization of crop parameters, which affects the credibility of the research results and needs to be further improved. The drought index method based on the crop model has been proposed and applied in many papers. The author needs to explain the improvement of the method in the paper. If there is no further improvement, it is necessary to mention the citation references. The manuscript lacks uncertainty analysis of this model and vulnerability assessment results, which reduces the credibility of the manuscript and needs further improvement. Minor comments: Page 3 Line 19-20: more explanation should be added for 'the third order' and 'the second order' Figure 2: Symbols should be explained in the legend Equation 2: How drought index is defined? Reference should be added. Page 7 Line 7-8: How are five levels of CLr defined? There is no detailed numbers. Page 8 Line 13-14: This sentence is not clear.

---

## Referee Comment (RC2) · Anonymous Referee #2 · 5 Dec 2019

The present manuscript aims at performing a crop-drought risk assessment in the European region for winter wheat using a crop drought vulnerability curve analysis. Vulnerability curves have been simulated thought EPIC model simulations using crop and meteorological data as input from $0.5°$ to 5' of gridded spatial resolution. The methodology proposed by using vulnerability curves with three key points and the clustering classification enabled a drought vulnerability assessment of winter wheat at the European region at a spatial scale. Other than an English proofing that prevented me from understanding some sentences properly, my main concerns rely on the lack of an uncertainty analysis enabling the reliability of the estimates to be assessed quantitatively. Furthermore, no details at all have been reported by the authors on the EPIC model,

neither in the literature experiences available to date.

---

## Author Comment (AC2) · 23 Mar 2020

Dear Anonymous Referee:

Thank you very much for your valuable comments, which are very helpful for our manuscript improving. We have studied the comments carefully and have made corrections which we hope to meet with approval.

The responses to your comments are structured in the sequence: (1) Point X: comments from Referees, (2) Response X: author's response, (3) Revision X: author's changes in manuscript (highlighted in red fonts). To better edit the formats for your

easy reading, we upload them as as a *.pdf file, which is displayed as a supplement to the comment. Once again, thank you very much for your comments and suggestions.

Yours sincerely,

Jing'ai Wang

Please also note the supplement to this comment:
https://www.nat-hazards-earth-syst-sci-discuss.net/nhess-2019-175/nhess-2019-175-AC2-supplement.zip

---

## Author Response (AR1)

**Author's Response**

We thank the Anonymous Referees for their comments about our research and we are pleased to find that our manuscript was carefully reviewed. All the comments are valuable and helpful for our manuscript improving. We have studied these comments carefully and have made corrections which we hope to meet with approval. Here are the specific responses to each Referee comment and the detailed modifications of the manuscript (marked in red).

**Response to Anonymous Referee #1**

***Point A-1: The English writing of the manuscript should be improved. Some sentences are hard to understand.***

**Response A-1:** We are very sorry for the understanding difficulties caused by our English writing. In order to solve this problem, we first carefully checked the full text and modified some paragraphs or sentences that may not be clear, especially the ones mentioned in the comments. On this basis, we asked a professional institute, American Journal Experts (AJE), for further English editing (verification code: A86C-32AC-F0EB-B41A-7FB9). All revision details can be found in the following marked-up manuscript.

***Point A-2: In addition, the definition of drought index is not clear. More explanation should be added.***

**Response A-2:** Thank you very much for your comments. We have elaborated on the definition of drought index as relative cumulative water stress during the crop growth period, which can reflect both water stress intensity and stress duration, and have added the reference for the calculation method (see Revision A-2-2). We have further supplemented the explanation of water stress in the manuscript, which is an output element of the EPIC model, reflecting the relationship between daily water supply and water demand (see Revision A-2-1 and A-2-2).

**Revision A-2-1 (section2.2, page 4, line 21-24 in the separate manuscript):**
In the process of simulation, intercepted photosynthetic active radiation is converted into potential biomass, which is adjusted by five daily stress factors (water, nitrogen, phosphorus, temperature, and aeration) to predict actual biomass growth, where the water stress (WS) factor is computed as the ratio of soil water use over potential plant water use.

**Revision A-2-2 (section 2.3.2, page 7, line34-38):**

As an output factor of the EPIC model, WS reflects the relationship between daily water supply and crop water demand. WS ranges from 0-1; the larger the value, the more serious the water shortage will be. To characterize integrated drought intensity affecting yield, drought intensity index (Di) is defined as relative cumulative water stress during the crop growth period, which can reflect both WS intensity and stress duration (Eq. (1), (2)) (Wang et al., 2013):

$$Di_i = \frac{HI_i}{\max(HI)} \; , \tag{1}$$

$$HI_i = \sum_{d=1}^{n}(WS_k) \; , \tag{2}$$

***Point A-3: The drought index method based on the crop model has been proposed and applied in many papers. The author needs to explain the improvement of the method in the paper. If there is no further improvement, it is necessary to mention the citation references.***

**Response A-3:** Thank you very much for your suggestions. We have added a reference (Wang et al., 2013) to the Drought Index method in the manuscript (see section 2.3.2, page 7, line38).

Wang, X. C., Li, J., Tahir, M. N., and Hao, M. D.: Validation of the EPIC model using a long-term experimental data on the semi-arid Loess Plateau of China, Mathematical and Computer Modelling, 54, 976-986,doi:10.1016/j.mcm.2010.11.025, 2011.

***Point A-4: There is no systematic introduction to the basic information of the Erosion-productivity Impact Calculator (EPIC) model or the research progress of this model in crop drought field.***

**Response A-4:** Thank you very much for your valuable comments. We have supplemented and improved the introduction to the basic information of the EPIC model (including the basic simulation mechanism) (see Revision A-4-1), the crop yield simulation research on the EPIC model in different water conditions (see Revision A-4-2), and the input and output data of the EPIC model in this manuscript (see Revision A-4-3). The above content shows that the EPIC model has good performance in yield simulation in a water stress environment, which supports our research well.

**Revision A-4-1 (section 2.2, page 4, line17-26):**

The EPIC model, published by the United States in 1984 (Williams et al., 1984), is selected to simulate the growth process of winter wheat. It can simulate soil erosion and productivity for hundreds of years on a daily step under a variety of climatic, environmental and management conditions. It simulates all crops with one model framework based on crop's physiological commonality and uses unique crop parameters for each crop. In the process of simulation,

intercepted photosynthetic active radiation is converted into potential biomass, which is adjusted by five daily stress factors (water, nitrogen, phosphorus, temperature, and aeration) to predict actual biomass growth, where the water stress (WS) factor is computed as the ratio of soil water use over potential plant water use. Crop yields are estimated as the product of the actual above ground biomass and a harvest index (economic yield/above ground biomass) (Williams et al., 1989).

**Revision A-4-2 (section 2.2, page 4, line27-page 5, line 6):**

EPIC model has been successfully applied in yield simulation for different crops and water input conditions in many parts of the world (Roloff et al., 1998;Gassman et al., 2005). Williams et al. (1989) described the EPIC model simulation results of 6 crop species throughout the U.S. and in European and Asian countries and concluded that the average simulated yields were always within 7% of the average measured yields. Bryant et al. (1992) used the EPIC model to duplicate 38 irrigation stress experiments in the Texas High Plains during 1975-1977 and found that simulated corn yields explained 83, 86, and 72 % of the variance in 3-year measured yields separately. Rinaldi (2001) simulated 66 irrigation scenarios for sunflower grown in Southern Italy, involving a combination of irrigation times, seasonal irrigation amounts and irrigation frequency, and obtained optimized irrigation scheduling without carrying out long and expensive field experiments. Ko et al. (2009) calibrated the EPIC model based on field studies in South Texas, and demonstrated that under full and deficient irrigation and rainfall conditions, EPIC-simulated yields of maize and cotton were in agreement with the measured yields according to a paired t-test. With good performance in water stress tests, EPIC model supports our research well.

**Revision A-4-3 (section 2.2, page 5, line 12-27):**

Inputs to EPIC include topography, soil, meteorological, and field management data. …
Outputs from the EPIC model include daily stress factors (water, nitrogen, phosphorus, temperature, and aeration) and annual yield value. The WS and yield can be further processed into sample for the construction of vulnerability curve.

*Point A-5: The time series of meteorological data is too short to prove the credibility of the vulnerability simulation results.*

**Response A-5:** Thank you very much for your valuable comments. These sections now describe meteorological data, including EPIC model calibration, model validation, and simulation of

water stress-scenario yield samples for vulnerability curve construction.

In terms of EPIC model calibration and validation, we originally used the meteorological data of 2000 for EPIC model calibration and the meteorological data of 2001-2004 for calibrated EPIC model validation. The time series of meteorological data was indeed relatively short, which could not adequately prove the credibility of the calibrated EPIC model in yield simulation. Therefore, we have supplemented the meteorological data of 1974-1999 (the meteorological data during this period is also processed based on the actual meteorological observation data) and obtained supplementary simulated yields based on calibrated EPIC to fully prove its credibility (see Revision A-5-1).

We re-compared the national-scale EPIC simulated yields and FAO statistical yields of 1974-2004 (excluding 2000 used for calibration). For national statistical yields with significant trends, linear de-trending transformations are applied to remove the impacts of technology progress (Xiong et al., 2014;Kamali et al., 2018a) (see Revision A-5-1). The R-square between the simulated and the statistical yields is 0.77, showing high consistency on a long-term scale and indicating reliable performance of EPIC for yield simulation. We have modified the data and methods sections and updated the results in Section 3.1 (see Revision A-5-2).

In terms of simulation of vulnerability samples, since the manuscript focuses on physical vulnerability, that is, the physical response of winter wheat to water shortage, we control the water supply condition using the meteorological data with suitable temperature and no precipitation and by setting a series of irrigation scenarios. Therefore, the reliability of the simulation results is not affected by the length of the processed ideal meteorological data. We have strengthened this description in the methods section of the article (see Revision A-5-3).

**Revision A-5-1: Supplementary validation time series in data and method sections**

**section 2.2, page 5, line 23-24:**

We use statistical yields of 2000 for model localization, and yields of other years between 1974 and 2004 for validation.

**section 2.3.1, page 7, line 18-22:**

To validate the parameterization results, we generate the simulated grid yields of 1974-2004 based on the calibrated EPIC model, and aggregate to the nation level by averaging. For FAO national statistical yields of 1974-2004 with significant trends, linear de-trending transformations are applied to remove the impacts of technology progress (Xiong et al., 2014;Kamali et al., 2018a). Then we compare national simulated yields with the statistical yields across all European countries.

**Revision A-5-2: Updated verification results (section 3.1, page 9, line 11-18)**

From the national comparison results from 1974 to 2004 (excluding calibration year of 2000),

though the simulated yields are slightly higher than the statistical yields, there is high agreement between the two (Fig. 3). The regression equation has an $R^2$ of 0.77 and passes the test with a confidence of 0.01, indicating a reliable performance of the calibrated EPIC model for yields simulation in various regions and various years.

[Figure]

**Figure 1: Comparison of national winter wheat yield reported by FAO and simulated by calibrated EPIC during the period from 1974-2004 (excluding calibration year of 2000).**

**Revision A-5-3: Simulation of physical vulnerability (section 2.3.2, page 7, line 28-32)**
To focus on physical drought vulnerability and eliminate the impact of other stress factors on yields, we use meteorological data with suitable temperature and no precipitation, and control the water supply condition by setting 20 irrigation scenarios, in which the irrigation amount uniformly increases from 0 to the optimum (the maximum irrigation amount without WS). The optimal value is determined by pre-testing. Consequently, we obtain the outputs of 20 groups of WS and yield for each grid evaluation unit.

*Point A-6: The spatial resolution of crop yield data is inconsistent with the spatial resolution of the simulation evaluation unit. It brings uncertainties to the validation of the model and the localization of crop parameters, which affects the credibility of the research results and needs to be further improved.*

**Response A-6:** Thank you very much for pointing out this issue, which is worthy of further discussion.
There are types of simulated evaluation units such as fields, sites, grids and countries, and the selection of them is usually related to the scale of the study area. For different types of simulation evaluation units, the available observed yield data will vary, resulting in different model calibration and validation methods (Wang et al., 2012), which can be roughly divided into the following two categories.

(1) Field or site-based simulation evaluation units are commonly used in small or medium-scale studies, where crop parameters usually apply the default values in the model, or relevant values in the literature or field experiments. Simulation evaluation units usually have the same parameters due to a small study area. Additionally, the model validation generally uses the field or site yield data obtained through field observation (Wang et al., 2011;Wang and Li, 2010;Ko et al., 2009;Sun et al., 2015;Cavero et al., 2000). Therefore, this type of study more easily achieves consistency between the spatial resolution of the observed yield data and the spatial resolution of the simulation evaluation unit.

(2) Grid-based simulation evaluation units are commonly used in large-scale studies, especially in continent or global-scale studies. When localizing crop parameters, some studies directly apply the default values in the model or relevant values in publications without considering spatial variability, due to the lack of crop variety distribution data on a large scale (Balkovič et al., 2013;Liu et al., 2007;Wriedt et al., 2009). Some studies perform partition calibration based on the natural environmental or administrative division. Each sub-region identifies a unique set of crop parameters, and all simulated evaluation units in one sub-region have the same crop parameters (Abbaspour et al., 2015;Angulo et al., 2013). For example, Kamali et al. (2018a) localized the crop parameters for each Sub-Saharan African country. They reiterated the simulation of maize yield on a grid at a spatial resolution of 0.5 °, aggregated to the national scale, and then compared to the FAO national statistical yields to obtain the optimal parameters. Xiong et al. (2014) divided the world into 433 homogenous units, selected 10 grids as calibration and validation units from each homogenous unit, and reiterated model simulations by incrementally changing the crop parameters and then compared them to the downscaled rice yield at 5' grid level resolution in International Food Policy Research Institute databases. Least Root Mean Square Error (RMSE) was chosen for the homogeneous unit. When validating the calibrated model, despite the method of calibration, large-scale studies generally aggregate the simulated grid yields to the administrative unit level to compare with the statistical yields (Balkovič et al., 2013;Liu et al., 2007;Kamali et al., 2018a;Xiong et al., 2014). Statistical yields used to estimate yield gap are generally based on yields reported in FAOSTAT and the Agro-MAPS project, a collaboration between FAO, IFPRI (International Food Policy Research Institute), SAGE (Centre for Sustainability and the Global Environment) and CIAT (The International Centre for Tropical Agriculture) (Ittersum et al., 2013).

In short, given the limited amount of data available, it is hard to ensure consistency between the spatial resolution of crop statistical yield data and the spatial resolution of the simulation evaluation unit on a large scale. To achieve model calibration and validation, it is necessary to aggregate the simulated grid yields to the statistical unit level or disaggregate statistical yields to the grid level.

This manuscript selects the European region as the study area and the 0.5 ° grid as the simulation evaluation unit. Therefore, we localize the parameters for each country based on the

idea of large-scale partition calibration. That is, we assume that all the grids in one country have the same winter wheat and then have the same set of crop parameters. We simply assign an FAO national statistical yield to the grids within a country, and this allocation method is consistent with Monfreda et al. (2008). The crop parameters are the optimal values for a country that minimize the RMSE between the simulated and assigned statistical grid yields. When validating the calibrated model, we aggregate the simulated grid yields to the country level by averaging and then compare the simulated and statistical yields across all countries.

Using such country-scale statistical yield data does bring some uncertainty to the localization of crop parameters and the validation of the model, but it is acceptable given the current situation with limited data. When more multi-year and higher-resolution statistical yield data is available in the future, the credibility of the results will be further improved. The above calibration and validation methods are further elaborated in the manuscript to avoid any unclear interpretation (see Revision A-6-1) and added to the discussion of model uncertainty to improve the explanation of limitations (see Revision A-6-2).

**Revision A-6-1 (section 2.3.1, page7, line 7-22):**

Considering the limitation of statistical yields on a grid scale, we localize the four key parameters at the country level based on the idea of partition calibration (Liu et al., 2007;Balkovič et al., 2013;Kamali et al., 2018a). That is, each country has a unique set of crop parameters, and all the grids within one country are the same. The default values of the crop parameters in the EPIC model are taken as the initial value, and the geographical environmental, field management and meteorological data are entered to obtain simulated grid yields of 2000. We simply assign a FAO national statistical yield to the grids within a country. Then the root mean square error (RMSE) between the simulated and statistical grid yields are calculated. We reiterate the yield simulations and RMSE calculations by incrementally adjusting the four key parameters to minimize RMSE. The calibration will be finished when the least RMSE is below the threshold or the number of reiterations is above the threshold.

To validate the parameterization results, we generate the simulated grid yields of 1974-2004 based on the calibrated EPIC model, and aggregate to the nation level by averaging. For FAO national statistical yields of 1974-2004 with significant trends, linear de-trending transformations are applied to remove the impacts of technology progress (Xiong et al., 2014;Kamali et al., 2018a). Then we compare national simulated yields with the statistical yields across all European countries.

**Revision A-6-2 (section 4.1, page 13, line14-20):**

The calibration and validation are carried out on the country level because of the limitation of available statistical yield data, which may cause some uncertainties for the input data. When more multi-year and higher-resolution statistical yield data are available in the future, the results

will be further improved. However, from the current comparison results of the national statistical yields and the simulated yields of the period from 1974-2004, the R2 of the two has reached 0.77, indicating a high reliability of the calibrated EPIC model.

***Point A-7: The manuscript lacks uncertainty analysis of this model and vulnerability assessment results, which reduces the credibility of the manuscript and needs further improvement.***

**Response A-7:** Thanks for the valuable suggestions. We have supplemented the uncertainty analysis of the model and vulnerability assessment results as the first section of the discussion according to your valuable suggestions. We discuss the uncertainties in two parts: (1) Uncertainties related to the EPIC model input parameters, including the selection of sensitive crop parameters, the limitation of available statistical yield data used for parameters calibration and validation. (2) Uncertainties related to the vulnerability assessment results that may be caused by vulnerability simulations and vulnerability curve constructions. The two types of uncertainties are evaluated quantitatively by comparing the national statistical and simulated yields of a relatively long period from 1974-2004, and by evaluating the standard deviations of reiterated vulnerability simulation results at sample grids selected randomly, respectively, by which the credibility of the manuscript are proved (see Revision A-7).

**Revision A-7 (section 4.1, page 13, line 6-page 14, line 23):**
The EPIC model default crop parameters may deviate from the actual growth in different regions, so we localize and verify the crop parameters to minimize these uncertainties. There are 56 crop parameters in the EPIC model, and different input parameters have different degrees of influence on the EPIC model in different simulation environments (Zhang et al., 2017). The main method to reduce the uncertainties of input parameters is to carry out sensitivity analysis in the basic evaluation unit and calibrate the sensitivity parameters one by one. However, this requires multiple calculations and does not completely eliminate the uncertainties of the input parameters(Yue et al., 2018). Therefore, with reference to previous research, we focus on the calibration and validation of the above four main sensitive parameters. The calibration and validation are carried out on the country level because of the limitation of available statistical yield data, which may cause some uncertainties for the input data. When more multi-year and higher-resolution statistical yield data are available in the future, the results will be further improved. However, from the current comparison results of the national statistical yields and simulated yields of the period from 1974-2004, the R2 of the two has reached 0.77, indicating a high reliability of the calibrated EPIC model.

To quantify the uncertainties of the vulnerability assessment results, we reiterate the vulnerability simulation and assessment 20 times and evaluate the standard deviation distribution of the results. First, we randomly select 10 % of samples from the five types of

vulnerability curves based on the principle of stratified sampling, and obtain a total of 201 sample grids. Next, according to the method in Section 2.3.1, we reiterate the vulnerability simulation and vulnerability curve construction process 20 times by changing the irrigation scenario settings, that is, keeping the non-irrigation and optimal irrigation scenarios unchanged and then randomly setting 18 irrigation scenarios between the two. From this, 20 reiterated vulnerability curves can be obtained for each sample grid. Then, by calculating the standard deviation of the loss rate for 20 reiterated vulnerability curves at the drought index interval of 0.1, the standard deviation of loss rate for each sample grid can be obtained to characterize the grid uncertainties. The mean standard deviation and 95 % prediction uncertainty band (95PPU) of total sample grids are finally calculated to characterize overall uncertainties. 95PPU is the range from 2.5 % to 97.5 % of the cumulative distribution function (Abbaspour et al., 2007). The results show that the mean standard deviation of loss rate is between 0 and 0.065, and the average is 0.033; the width of PPU95 is between 0.007 and 0.135, and the average is 0.067; the two indicators reach the peak when the drought index is between 0.4 and 0.7 (Fig. 7). Although the prediction uncertainty of loss rate is relatively large in such range, it is still significantly smaller than the difference in loss rate between regions (which can reach more than 0.5), so it has little effect on the distribution pattern of vulnerability. In summary, the vulnerability assessment results of this paper are credible.

[Figure]

**Figure 7: Distribution of standard deviation of loss rate under different drought index. The mean standard deviation and 95 % prediction uncertainty band (95PPU) are calculated by the standard deviations of sample grids, which are randomly selected from the five vulnerability curves at a proportion of 10 %.**

*Point A-8: The exposition of the theory and practice progress regarding crop drought vulnerability is insufficient in this research. It needs to be supplemented with a large amount of literature, especially those in the past 15 years, to discuss the historical research process of vulnerability assessment, which is from the single index to the linear index, and then to the curve index.*

**Response A-8:** Thank you very much for your valuable comments. We have supplemented some recent literature and relevant research cases and made key revisions to the introduction

based on your comments. According to the definition of vulnerability of the United Nations International Strategy for Disaster Reduction (UNISDR), vulnerability is the characteristics and circumstances of an affected body that make it susceptible to the damaging effects of a hazard (UNISDR, 2009). We have reclassified vulnerability research into three categories in the manuscript on the basis of different research methods. (1) Calculation of the vulnerability index based on selected relevant indicators. The characteristic of this method is to establish the evaluation index system and weight system based on meteorological, hydrogeological and other factors to form a vulnerability evaluation (Wilhelmi and Wilhite, 2002;Pandey et al., 2010;Simelton et al., 2009). (2) Quantitative research on vulnerability based on historical statistics and meteorological observations. This method mainly uses meteorological observation data, historical statistical data and so on to establish the quantitative relationship between disaster intensity and historical disaster loss through a regression model (Xu et al., 2013;Jayanthi et al., 2014;Fishman, 2016). (3) Quantitative research on vulnerability curves based on field experiments and crop model simulations. The method is characterized by conducting field control experiments or crop growth model simulations through artificially setting up different disaster intensity scenarios and then fitting vulnerability curves from the perspective of the crop disaster-causing mechanism (Wang et al., 2013;Yin et al., 2014;Kamali et al., 2018b). The above three aspects, vulnerability assessment by index construction and by regression model, and vulnerability curve construction based on the disaster-causing mechanism, have been expanded in the manuscript (see Revision A-8).

[revised manuscript text omitted]

***Minor comment A-a: Page 3 Line 19-20: more explanation should be added for 'the third order' and 'the second order'.***

**Response A-a:** We are sorry for the unclear expression before and we correct 'the second/third order' to 'the second/third derivative of the vulnerability curve':

"Therefore, the key points of the vulnerability curve—the transition points of three stages (P1 and P3, where the third derivative of the vulnerability curve is equal to zero) and the turning point of the loss growth rate (P2, where the second derivative of the vulnerability curve equals zero ) are used to describe the loss change characteristics, the cumulative loss to the loss extent characteristics, and the morphological classification to of the integrated description."

***Minor comment A-b: Figure 2: Symbols should be explained in the legend.***

**Response A-b:** Thanks for the suggestion. We have supplemented the explanation of the symbols in Figure 2:

[Figure]

**Figure 2: Basic research framework. First, we input relevant data into the EPIC model and perform model calibration. Next, we obtain a series of water stress and yield data based on the calibrated EPIC model by setting different irrigation scenarios, which are converted into drought index (Di) and yield loss rate (Lr) for the construction of vulnerability curves. Then, we extract three key points and calculate the cumulative loss rate of vulnerability curves for the spatial analysis of loss change and loss extent characteristics. Finally, we calculate the Lr and the growth rate of Lr (Lr') under a set of fixed Di to transform the vulnerability curves into a finite data set for clustering, and the classification of vulnerability curves can be used for the integrated spatial analysis.**

*Minor comment A-c: Equation 2: How drought index is defined? Reference should be added.*

**Response A-c:** Thank you for your comments. We have supplemented the definition of drought index with references to its calculation formulas. More details can be found in Responses 2 and 3 above.

*Minor comment A-d: Page 7 Line 7-8: How are five levels of CLr defined? There is no detailed numbers.*

**Response A-d:** Thank you for your suggestion. We have supplemented detailed numbers of five levels of CLr:

"All CLr values are divided into five levels by the natural breakpoint method: extremely low (0.22-0.34), low (0.34-0.42), moderate (0.42-0.49), high (0.49-0.55), and extremely high (0.55-0.69)."

*Minor comment A-e: Page 8 Line 13-14: This sentence is not clear.*

**Response A-e:** Thanks for your comment and sorry for our negligence on the English writing. We have modified this sentence and its related context to make it clear:

"For most grids, the Di values at the three key points are mainly distributed from 0.15-0.55, 0.35-0.7 and 0.4-0.8, while the Lr values have a relatively small distribution, from 0.1-0.2, 0.4-0.5 and 0.7-0.8. Therefore, the characteristics of stage transitions of grid vulnerability curves can be simplified by only using the Di at key points instead of two coordinates. The larger the Di is at key points, the more severe the drought must be to cause a similar loss rate; this is

reflected in the lag in the stage transitions of vulnerability curve, indicating a greater tolerance to drought disturbance."

**Response to Anonymous Referee #2**

***Point B-1: Other than an English proofing that prevented me from understanding some sentences properly, …***

**Response B-1:** We are very sorry for the understanding difficulties caused by English writing and have tried our best to improve them. We first carefully reviewed and revised the writing of the entire manuscript, and then we asked a professional institute, American Journal Experts (AJE), for further English editing (verification code: A86C-32AC-F0EB-B41A-7FB9). All revision details can be found in the following marked-up manuscript.

***Point B-2: …, my main concerns rely on the lack of an uncertainty analysis enabling the reliability of the estimates to be assessed quantitatively.***

**Response B-2:** Thank you very much for your valuable suggestions. We have supplemented the uncertainty analysis as the first section of discussion, which includes two parts:

(1) Uncertainties related to the EPIC model input parameters, including the selection of sensitive crop parameters, the limitation of available statistical yield data used for parameters calibration and validation. For better quantitative evaluation of calibration results, we have extended the time series of simulated and statistical yield data for comparison, that is, from 2001-2004 to 1974-2004 (see Revision B-2-1). According to new comparison results, the $R^2$ of the two has reached 0.77, indicating a high reliability in the calibrated EPIC model (see Revision B-2-2 and B-2-3).

(2) Uncertainties related to the vulnerability assessment results. For assessing quantitatively, we reiterate the vulnerability simulation and vulnerability curve constructions 20 times at randomly selected grids, the standard deviation of loss rate in 20 repeated vulnerability curves of each sample grid is obtained to characterize the grid uncertainties, and the mean standard deviation and 95 % prediction uncertainty band (95PPU) of loss rate of total are calculated characterize overall uncertainties. The results show that the mean standard deviation of loss rate is between 0 and 0.065; the width of PPU95 is between 0.007 and 0.135; and the two indicators reach the peak when the drought index is between 0.4 and 0.7. Although the prediction uncertainty of loss rate is relatively large in such range, it is still significantly smaller than the difference in loss rate between regions, so it has little effect on the distribution pattern of vulnerability. In summary, the vulnerability assessment results of this paper are credible (see Revision B-2-3).

**Revision B-2-1: Supplementary validation time series in data and method sections**

**section 2.2, page 5, line 23-24:**

[revised manuscript text omitted]

*Point B-3: Furthermore, no details at all have been reported by the authors on the EPIC model, neither in the literature experiences available to date.*

**Response B-3:** Thank you very much for your valuable comments. We have make supplements and modifications to Section 2.2 in the manuscript, which elaborate the basic information of the EPIC model and the crop yield simulation researches of the EPIC model in different water conditions. The above content shows that the EPIC model has good performance in yield simulation under water stress environment, which supports our research well (see Revision B-3).

**Revision B-3 (section 2.2, page 4, line17- page 5, line 6):**
The EPIC model, published by the United States in 1984 (Williams et al., 1984), is selected to simulate the growth process of winter wheat. It can simulate soil erosion and productivity for hundreds of years on a daily step under a variety of climatic, environmental and management conditions. It simulates all crops with one model framework based on crop's physiological commonality and uses unique crop parameters for each crop. In the process of simulation, intercepted photosynthetic active radiation is converted into potential biomass, which is adjusted by five daily stress factors (water, nitrogen, phosphorus, temperature, and aeration) to predict actual biomass growth, where the water stress (WS) factor is computed as the ratio of soil water use over potential plant water use. Crop yields are estimated as the product of the actual above ground biomass and a harvest index (economic yield/above ground biomass) (Williams et al., 1989).

EPIC model has been successfully applied in yield simulation for different crops and water input conditions in many parts of the world (Roloff et al., 1998;Gassman et al., 2005). Williams

et al. (1989) described the EPIC model simulation results of 6 crop species throughout the U.S. and in European and Asian countries and concluded that the average simulated yields were always within 7% of the average measured yields. Bryant et al. (1992) used the EPIC model to duplicate 38 irrigation stress experiments in the Texas High Plains during 1975-1977 and found that simulated corn yields explained 83, 86, and 72 % of the variance in 3-year measured yields separately. Rinaldi (2001) simulated 66 irrigation scenarios for sunflower grown in Southern Italy, involving a combination of irrigation times, seasonal irrigation amounts and irrigation frequency, and obtained optimized irrigation scheduling without carrying out long and expensive field experiments. Ko et al. (2009) calibrated the EPIC model based on field studies in South Texas, and demonstrated that under full and deficient irrigation and rainfall conditions, EPIC-simulated yields of maize and cotton were in agreement with the measured yields according to a paired t-test. With good performance in water stress tests, EPIC model supports our research well.

**References in the point-by-point response**

1. Abbaspour, K. C., Rouholahnejad, E., Vaghefi, S., Srinivasan, R., Yang, H., and Kløve, B.: A continental-scale hydrology and water quality model for Europe: Calibration and uncertainty of a high-resolution large-scale SWAT model, Journal of Hydrology, 524, 733-752,doi:10.1016/j.jhydrol.2015.03.027, 2015.

2. Angulo, C., Rötter, R., Lock, R., Enders, A., Fronzek, S., and Ewert, F.: Implication of crop model calibration strategies for assessing regional impacts of climate change in Europe, Agricultural and Forest Meteorology, 170, 32-46,doi:10.1016/j.agrformet.2012.11.017, 2013.

3. Balkovič, J., van der Velde, M., Schmid, E., Skalský, R., Khabarov, N., Obersteiner, M., Stürmer, B., and Xiong, W.: Pan-European crop modelling with EPIC: Implementation, up-scaling and regional crop yield validation, Agricultural Systems, 120, 61-75,doi:10.1016/j.agsy.2013.05.008, 2013.

4. Barros, I. d., Williams, J. R., and Gaiser, T.: Modeling soil nutrient limitations to crop production in semiarid NE of Brazil with a modified EPIC version: II: Field test of the model, Ecological Modelling, 181, 567-580,doi:https://doi.org/10.1016/j.ecolmodel.2004.03.018, 2005.

5. Cavero, J., Farre, I., Debaeke, P., and Faci, J. M.: Simulation of Maize Yield under Water Stress with the EPICphase and CROPWAT Models, Agronomy Journal, 92, 679,doi:10.2134/agronj2000.924679x, 2000.

6. Fishman, R.: More uneven distributions overturn benefits of higher precipitation for crop yields, Environmental Research Letters, 11, 024004,doi:10.1088/1748-9326/11/2/024004, 2016.

7. Ittersum, M. K. v., Cassman, K. G., Grassini, P., Wolf, J., Tittonell, P., and Hochman, Z.: Yield gap analysis with local to global relevance—A review, Field Crops Research, 143, 4-17,doi:10.1016/j.fcr.2012.09.009, 2013.

8. Jayanthi, H., Husak, G. J., Funk, C., Magadzire, T., Adoum, A., and Verdin, J. P.: A probabilistic approach to assess agricultural drought risk to maize in Southern Africa and millet in Western Sahel using satellite estimated rainfall, International Journal of Disaster Risk Reduction, 10, 490-502,doi:10.1016/j.ijdrr.2014.04.002, 2014.

9. Kamali, B., Abbaspour, K. C., Lehmann, A., Wehrli, B., and Yang, H.: Uncertainty-based auto-calibration for crop yield – the EPIC+ procedure for a case study in Sub-Saharan Africa, European Journal of Agronomy, 93, 57-72,doi:10.1016/j.eja.2017.10.012, 2018a.

10. Kamali, B., Abbaspour, K. C., Lehmann, A., Wehrli, B., and Yang, H.: Spatial assessment of maize physical drought vulnerability in sub-Saharan Africa: Linking drought exposure with crop failure, Environmental Research Letters, 13, 074010,doi:10.1088/1748-9326/aacb37, 2018b.

11. Ko, J., Piccinni, G., Guo, W., and Steglich, E.: Parameterization of EPIC crop model for simulation of cotton growth in South Texas, The Journal of Agricultural Science, 147, 169-178,doi:10.1017/s0021859608008356, 2009.

12. Liu, J., Williams, J. R., Zehnder, A. J. B., and Yang, H.: GEPIC – modelling wheat yield and crop water productivity with high resolution on a global scale, Agricultural Systems, 94, 478-493,doi:10.1016/j.agsy.2006.11.019, 2007.

13. Monfreda, C., Ramankutty, N., and Foley, J. A.: Farming the planet: 2. Geographic distribution of crop areas, yields, physiological types, and net primary production in the year 2000, Global Biogeochemical Cycles, 22,doi:10.1029/2007gb002947, 2008.

14. Pandey, R. P., Pandey, A., Galkate, R. V., Byun, H.-R., and Mal, B. C.: Integrating Hydro-Meteorological and Physiographic Factors for Assessment of Vulnerability to Drought, Water Resources Management, 24, 4199-4217,doi:10.1007/s11269-010-9653-5, 2010.

15. Simelton, E., Fraser, E. D. G., Termansen, M., Forster, P. M., and Dougill, A. J.: Typologies of crop-drought vulnerability: an empirical analysis of the socio-economic factors that influence the sensitivity and resilience to drought of three major food crops in China (1961–2001), Environmental Science & Policy, 12, 438-452,doi:10.1016/j.envsci.2008.11.005, 2009.

16. Sun, Z., Zhang, J., Yan, D., Wu, L., and Guo, E.: The impact of irrigation water supply rate on agricultural drought disaster risk: a case about maize based on EPIC in Baicheng City, China, Natural Hazards, 78, 23-40,doi:10.1007/s11069-015-1695-9, 2015.

17. UNISDR: 2009 UNISDR terminology on disaster risk reduction, Geneva: UNISDR, 2009.

18. Wang, X., Williams, J. R., Gassman, P. W., Baffaut, C., Izaurralde, R. C., Jeong, J., and Kiniry, J. R.: EPIC and APEX: Model Use, Calibration, and Validation, Transactions of the Asabe, 55, 1447-1462,doi:10.13031/2013.42253, 2012.

19. Wang, X. C., and Li, J.: Evaluation of crop yield and soil water estimates using the EPIC model for the Loess Plateau of China, Mathematical and Computer Modelling, 51, 1390-1397,doi:10.1016/j.mcm.2009.10.030, 2010.

20. Wang, X. C., Li, J., Tahir, M. N., and Hao, M. D.: Validation of the EPIC model using a long-term experimental data on the semi-arid Loess Plateau of China, Mathematical and Computer Modelling, 54, 976-986,doi:10.1016/j.mcm.2010.11.025, 2011.

21. Wang, Z., He, F., Fang, W., and Liao, Y.: Assessment of physical vulnerability to agricultural drought in China, Natural Hazards, 67, 645-657,doi:10.1007/s11069-013-0594-1, 2013.

22. Wilhelmi, O. V., and Wilhite, D. A.: Assessing Vulnerability to Agricultural Drought: A Nebraska Case Study, Natural Hazards, 25, 37-58,doi:10.1023/a:1013388814894, 2002.

23. Williams, J., Jones, C., Kiniry, J., and Spanel, D.: The EPIC crop growth model, Trans. ASAE, 32, 497-511, 1989.

24. Wriedt, G., Van der Velde, M., Aloe, A., and Bouraoui, F.: Estimating irrigation water requirements in Europe, Journal of Hydrology, 373, 527-544,doi:10.1016/j.jhydrol.2009.05.018, 2009.

25. Xiong, W., Balkovič, J., van der Velde, M., Zhang, X., Izaurralde, R. C., Skalský, R., Lin, E., Mueller, N., and Obersteiner, M.: A calibration procedure to improve global rice yield simulations with EPIC, Ecological Modelling, 273, 128-139,doi:10.1016/j.ecolmodel.2013.10.026, 2014.

26. Xu, X., Ge, Q., Zheng, J., Dai, E., Zhang, X., He, S., and Liu, G.: Agricultural drought risk analysis based on three main crops in prefecture-level cities in the monsoon region of east China, Natural Hazards, 66, 1257-1272,doi:10.1007/s11069-012-0549-y, 2013.

27. Yin, Y., Zhang, X., Lin, D., Yu, H., Wang, J. a., and Shi, P.: GEPIC-V-R model: A GIS-based tool for regional crop drought risk assessment, Agricultural Water Management, 144, 107-119,doi:10.1016/j.agwat.2014.05.017, 2014.

28. Yue, Y., Wang, L., Li, J., and Zhu, A. x.: An EPIC model-based wheat drought risk assessment using new climate scenarios in China, Climatic Change,doi:10.1007/s10584-018-2150-1, 2018.

29. Zhang, X., Guo, H., Wang, R., Lin, D., Gao, Y., Lian, F., and Wang, J. a.: Identification of the Most Sensitive Parameters of Winter Wheat on a Global Scale for Use in the EPIC Model, Agronomy Journal, 109, 58-70,doi:10.2134/agronj2016.06.0347, 2017.

**List of all relevant changes made in the manuscript**

- The affiliations of co-authors were updated.
- Some sentences were modified to be clearer.
- The research progress in the introduction was expanded and rewritten.
- The basic information and the related applied researches of the EPIC model were supplemented.
- Table1: The description of administrative boundary data was supplemented, and the years of historical daily meteorological data and statistical yield data were updated.
- Figure 2: The explanation of the symbols was supplemented.
- The calibration and validation methods of the EPIC model were modified.
- More explanation on the definition of drought index and the related citation reference were supplemented.
- Section 3: The validation results of the EPIC model (including Figure 3 and text) were updated.
- Uncertainty analysis of the EPIC model and vulnerability simulation results was added to the discussion.
- The references were updated.

[revised manuscript text omitted]

---

## Author Response (AR3)

**Author's Response**

We thank all the Anonymous Referees for their comments about our research and we are pleased to find that our manuscript was carefully reviewed. All the comments are valuable and helpful for our manuscript improving. We have studied these comments carefully and have made corrections which we hope to meet with approval. Here are the specific responses to each Referee comment and the detailed modifications of the manuscript.

**Report #1 by Referee #2**

Referees #1 suggested to accept the manuscript, without new revision comments. We are very grateful for his/her last comments on 05 Dec 2019, which greatly contributed to the improvement of our manuscript.

**Report #2 by Referee #1**

***Point 1: The article is lack of systematic analysis to the application research progress of this model in crop drought assessment field.***

**Response 1:** Thanks to referee for pointing out the above issues. We have revised and supplemented the analysis on the application research progress of the EPIC model (Please refer to page 5, line 3-22). We first review its simulation performance under different water stress environments, which shows that the model has good crop yield simulation capabilities (Bryant et al., 1992;Ko et al., 2009). On this basis, we further introduced its application in crop drought assessment field, including irrigation management (Rinaldi, 2001), drought impact prediction (Webber et al., 2018;Leng and Hall, 2019) and drought vulnerability assessment (Wang et al., 2013;Kamali et al., 2018c). It is found that the model can effectively provide fine yield loss data for drought assessment by inputting drought scenario data, which means it can be a good technical support for our research.

***Point 2: The author should state how to choose the curve of vulernability assessment, rather than a straight line.***

**Response 2:** We are very grateful to the referee for his/her valuable comments. The vulnerability curve describes the functional relationship between drought intensity and loss. As drought intensify, disaster losses begin to appear and gradually increase until the end of the disaster. That is regarded as an interactive process of energy accumulation and resisting effect

(Chen et al., 2015;Chen et al., 2017). Drought intensification brings about energy accumulation, which will be released when it reaches a certain level; meanwhile, resistance, such as system adjustment ability, always exists. In the initial stage, it appears as a slow development of drought due to insufficient energy storage and the existence of resistance. And if the driving force is stopped or weakened, the energy accumulation basically ends. Otherwise, energy will continue to accumulate, then break through the resistance and release, resulting in explosive development. Finally, the drought event gradually subsided with energy attenuation and resistance influence.

The linear trend cannot well describe the above-mentioned beginning and end of disaster changes, and it tends to represent that the disaster growth will not end. Therefore, we argue that the relationship between drought intensity and loss is a non-linear, monotonically increasing function, and has at least two critical points representing the initial, development and attenuation stages change. These characteristics are consistent with the S-shaped curve. So we select the typical logistics model in the S-shaped curves for vulnerability curve construction (Skrobacki, 2007). We have modified the Section 2.1 of the manuscript to better illustrate this point (page 3, line 34-Page 4, line6).

***Point 3: The author needs to explain what is the improvement over previous research about the drought index method. If there is no further improvement, the author needs to state the innovative content of the article.***

**Response 3:** We thank the referee for putting forward the constructive comment. The vulnerability assessment methods over previous research can be divided into three categories: the index method based on selected relevant indicators, the statistical method based on historical disaster data, and the vulnerability curve method based on based on field experiments and crop model simulations. However, the index method can only express the relative level of vulnerability between regions, but cannot quantitatively predict the loss(Wilhelmi and Wilhite, 2002;Simelton et al., 2009;Wu et al., 2010); the statistical method is easily affected by the availability and quality of disaster loss data, and is difficult to apply to high-resolution spatial analysis(Lobell and Burke, 2008;Hlavinka et al., 2009;Rowhani et al., 2011); the vulnerability curve belongs to infinite dimensional data and is difficult to conduct spatial analysis directly, mainly used in risk assessment field with insufficient vulnerability information mining (Kamali et al., 2018a;Yin et al., 2014).

In this context, the main innovative content of this article is to putting forward the vulnerability curve feature extraction and spatial difference analysis method, which improves the quantitative degree of vulnerability spatial analysis. In order to highlight such innovation, we emphasize the limitations of previous research in the introduction. Please refer to line 18 at page 2 to line 24 at page 3.

***Point 4: The study needs to explain how to ensure classification method of drought index.***

**Response 4:** Thanks for giving us this effective suggestion. Actually, the vulnerability curve contains indicators in two dimensions: drought index and loss rate when performing spatial analysis. We classify the vulnerability curve according to the attributes of these two dimensions. We referred to the idea of general curve clustering when clustering the vulnerability curve (James and Sugar, 2003). The first step is to filter the infinite dimensional curve data to a finite set of representative parameters. In order to represent the loss and loss change characteristics of the S-shaped vulnerability curve comprehensively, we choose the loss rate and the growth rate of loss rate under fixed drought index as representative parameters.

The second step is to select an appropriate clustering tool for the representative parameters. K-means is a clustering algorithm which based on partition. It has the characteristics of faster calculation speed and good clustering effect; moreover, it has been widely used in clustering analysis (Sun et al., 2008). We utilized the Euclidean distance to compare the similarity of vulnerability curve among grid cells (Jacques et al., 2014). The smaller the distance, the more similar the vulnerability curves. The selection of optimal K value is the key to K-means clustering, which was determined by the elbow method and the density of each cluster comprehensively (Nainggolan et al., 2019;Wang et al., 2019). We add more details in the section 2.3.2. (page 9, line 22- page 10, line 26) and section 3.3 (page 13, line 5-11).

***Point 5: The evaluation unit of national statistical yield data and drought index grid data(0.5°×0.5°) is not matched. So, the credibility of the article is lack.***

**Response 5:** Thanks to the referee for putting forward this valuable comment, which is worthy of discussion. The drought index data and simulated yield data (grid unite, $0.5°\times0.5°$) were obtained through the EPIC model simulation. In the process, muti-year statistical and simulated yield data is needed to calibrate crop parameters and validate simulate results. However, unlike studies under site and field scale(Wang et al., 2011;Wang and Li, 2010;Ko et al., 2009;Sun et al., 2015;Cavero et al., 2000), it is difficult to obtain the observational yield data of all grid units for many years on continent scale. Therefore, **limited by the availability of data, the inconsistence between the spatial resolution of crop statistical yield data and the spatial resolution of the simulation evaluation unit on a large scale is a common problem.**

In response to the lack of data, some studies directly apply the default values in the EPIC model or relevant values in publications, assuming that the crop parameters in the region are homogeneous and then avoiding using statistical yield for calibration (Balkovič et al., 2013;Liu et al., 2007;Wriedt et al., 2009). Some studies perform partition calibration based on the natural environmental or administrative division, based on the assumption that the crop parameters in a sub-region are homogeneous, and identify the unique crop parameters of each sub-region (Abbaspour et al., 2015;Angulo et al., 2013). Then, they assign national statistical yield data to each grid within the country. When simulated yield of grids are generally closest to the country

statistical yield, the optimal crop parameters are obtained (Kamali et al., 2018b). In terms of model validation, it is common to aggregate the simulated grid yields to the national-scale level for comparison(Xiong et al., 2014;Abbaspour et al., 2015;Kamali et al., 2018b) . **In short, national statistical yield is a commonly used data source in larger-scale studies** (Ittersum et al., 2013). So we applied the above-mentioned partition calibration and up-scaling validation method.

We acknowledge that the spatial resolution unconsistency between the two types of data will bring certain uncertainty to the crop parameters localization and validation, **and we have raised this point in the discussion section (page 17, line 9-13).** However, it is acceptable under current situation with limited data. When more multi-year and higher-resolution statistical yield data is available in the future, the calibration and validation of model will be further improved.

***Point 6:   This paper needs to analyze the shortcomings of this study and the research direction of the next step.***

**Response 6:** We appreciate and agree with this constructive suggestion. We have revised the section of discussion as follows. The section 4.2 mainly discusses the uncertainty and limitations of two aspects, the EPIC model calibration, and the vulnerability simulation assessment (page 16, line 32-page 18, line 4). The uncertainty and limitations of the calibration mainly include the selection of calibrated crop parameters and the accuracy of the statistical yield data; the uncertainty in the simulation evaluation process may mainly come from the experimental design, so we conducted quantitative analysis by changing the irrigation scenario settings and repeating the experiment.

On the basis of introducing the application value of the vulnerability curve, section 4.3 further proposed two major directions for future research. One is to conduct a comprehensive vulnerability assessment combined with social vulnerability, and the other is to develop dynamic vulnerability evaluation through considering climate change and socio-economic changes. Please refer to page 18, line 5-page 19, line 8 for more details.

**Report #3 by Referee #3**

***Point 7:   I have some concerns about the model that they are applying but it is one way to do it. However, in the vulnerability curve fit there are very few information and they should detail a little more.***

**Response 7:**   We really appreciate these constructive comments. We further explained the reasons why we chose the S-shaped curve for fitting from the perspective of the interaction between energy accumulation and resisting effect. Please refer to the **Response 2.**

Regarding the effect of vulnerability curve fit, we use coefficient of determination ($R^2$) and

Root Mean square error (RMSE) to measure (Quiring and Papakryiakou, 2003). We emphasized this part in the method, referring to page 8, line 29-33. Then we analyse these two indicators from the grid and regional scale at page 11, line 9-16. The results show a good goodness of fit.

*Point 8:  About the clustering, the authors claim they applied k-means clustering but even in this point you have several options to chose and they don't specify which options and why the have chosen it. Part of the appendix should be included in the text.*

*Just realise that the k-means is based in several parameters. Which is the number of clusters? how do you chose that number? This is an important point as it is the core of this research.*

**Response 8:** Thanks for the valuable suggestions on K-means clustering and we fully agree.

To explain why we chose this cluster algorithm, we have added explanation of clustering analysis algorithms and K-means clustering algorithm in the methods section (Please refer to line 22 at page 9 to line 1 at page 10). K-means belongs to partition clustering algorithm. It has the characteristics of faster calculation speed and excellent clustering effect, and it is the most widely method of clustering analysis (Han et al., 2012;Sun et al., 2008). Based on such reasons, we decide to choose it as the clustering tool.

In the methods section, we also add the introduction of the algorithm principle and the parameters selection of K-means clustering (Please refer to lines 1-26 at page 10). The classical K-means uses Euclidean distance to compare the similarity of data points, and classifies data through multiple iterations. The number of clusters needs to be set in advance, which is the key to the algorithm. In order to obtain the optimal value, we use the commonly used elbow method, combining the density of each cluster (Nainggolan et al., 2019;Wang et al., 2019). This part was originally expressed by Appendix B, and it has now been modified into the text to better support the research results. Please refer to line 5-11 at page 13.

*Point 9:  Some details. The legends for graphs and tables should be improved.*

**Response 9:** We are sorry for the imperfections in details. During the last two responses, we repeatedly checked all the graphs and tables in this article, and made the corresponding modifications to improve them as much as possible, such as adding coordinates and legends (Fig.3, Table 4), modifying the size of legend symbols and text (Fig.4, Fig.5, Fig.6, Fig 8), standardizing serial numbers (Fig.1, Fig.4, Fig.5, Fig.6, Fig 8) and so on.

**List of all relevant changes made in the manuscript**

- The affiliations of co-authors were updated.
- The introduction is revised to highlight the innovation.
- The related applied researches of the EPIC model are supplemented.
- The reasons for choosing the S-shaped vulnerability curve are further supplemented.
- Application research progress of EPIC model in crop drought assessment field are supplemented.
- The fitting result of the vulnerability curves is supplemented.
- More explanations on K-means clustering method are added.
- Research limitations and prospects are supplemented.
- The calibration and validation methods of the EPIC model were modified.
- Some figures, tables and appendices are modified.
- Some references are supplemented.